# LTSM-Bundle: A Toolbox and Benchmark on Large Language Models for Time Series Forecasting

## Abstract

Time Series Forecasting (TSF) has long been a challenge in time series analysis. Inspired by the success of Large Language Models (LLMs), researchers are now developing Large Time Series Models (LTSMs), universal transformer-based models that use autoregressive prediction to improve TSF. However, training LTSMs on heterogeneous time series data poses unique challenges, including diverse frequencies, dimensions, scalability, and patterns across datasets. Recent efforts have studied and evaluated various design choices aimed at enhancing LTSM training and generalization capabilities. Despite progress in both paradigms, there is no unified framework for systematically evaluating models and design choices across them. However, these design choices are typically studied and evaluated in isolation and are not compared collectively. In this work, we introduce `LTSM-Bundle`, a comprehensive toolbox and benchmark for training LTSMs, spanning pre-processing techniques, model configurations, and dataset configurations. Modularized and benchmarked LTSMs from multiple dimensions, encompassing prompting strategies, tokenization approaches, training paradigms, base model selection, data quantity, and dataset diversity. By combining the most effective design choices, the combination achieves state-of-the-art zero-shot and few-shot performance while providing a reproducible foundation for evaluating both traditional LSF models and emerging LTSMs. Our source code is available at `https://anonymous.4open.science/r/LTSM-bundle-5B70/`

## 1 Introduction

Time series forecasting (TSF) is a long-standing task in time series analysis, aiming to predict future values based on historical data points. Over the decades, TSF has transitioned from traditional statistical methods (Ariyo et al., 2014) to machine learning (Friedman, 2001), and more recently, to deep learning approaches (Elsworth & Güttel, 2020; Livieris et al., 2020). In particular, transformers (Vaswani et al., 2017), which are often regarded the most powerful architecture for sequential modeling, have demonstrated superior performance in TSF, especially for long-term forecasting (Wu et al., 2021; Zhou et al., 2021; Nie et al., 2022; Woo et al., 2022; Kitaev et al., 2020). Moving forward, inspired by the remarkable capabilities of Large Language Models (LLMs), many researchers have begun to explore Large Time Series Models (LTSMs) as the natural next phase, seeking to train universal transformer-based models for TSF (Woo et al., 2024; Garza & Mergenthaler-Canseco, 2023; Dooley et al., 2024; Rasul et al., 2024; Das et al., 2023; Gruver et al., 2024; Chang et al., 2023; Zhou et al., 2023; Jin et al., 2023).

Unlike textual data, where tokens typically hold semantic meanings transferable across documents, time series data exhibits high heterogeneity, presenting unique challenges for LTSM training. Across different datasets, time series often have diverse frequencies (such as hourly and daily), dimensions (in terms of varying numbers of variables) and patterns (where, for example, traffic time series may differ significantly from electricity data). This diversity not only poses difficulties in training an LTSM to fit all the datasets but also impedes the model's generalization to unseen data.

Recent endeavors have proposed various innovative designs to enhance the training and generalization capability of LTSMs. To name a few, (i) in terms of pre-processing, prompting strategies have been proposed

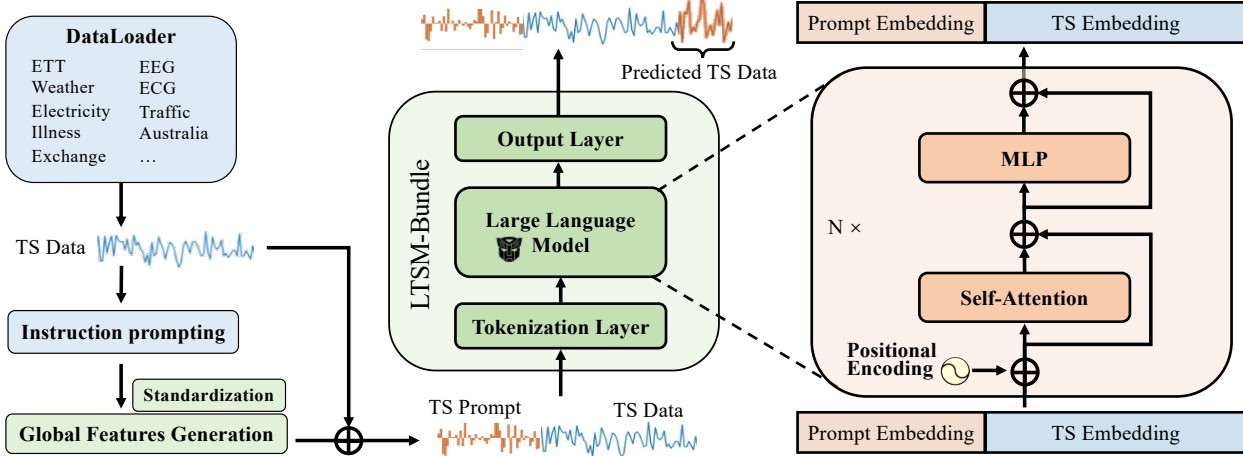

Figure 1: An overview of the key design choices supported by the `LTSM-Bundle` framework. The library is a general-purpose benchmarking toolbox that standardizes data preprocessing, tokenization strategies, training paradigms, and evaluation pipelines across traditional LSF models, LLM-based models, and pre-trained LTSMs, rather than being limited to LLM-centric designs.

to generate dataset-specific prompts (Jin et al., 2023), while various tokenization strategies have been studied for converting time series into tokens to be inputted into transformer layers (Zhou et al., 2023; Ansari et al., 2024); (ii) for the model configurations, prior research has involved reusing weights from pre-trained language models and adapting them to downstream tasks (Zhou et al., 2023); (iii) regarding dataset configurations, different datasets have been utilized for training purposes (Garza & Mergenthaler-Canseco, 2023; Zhou et al., 2023; Chang et al., 2023). Despite these advances, evaluating models across both paradigms remains challenging. Existing studies often adopt inconsistent pre-processing pipelines, heterogeneous tokenization strategies, and non-uniform evaluation metrics, making it difficult to draw fair comparisons or identify best practices, where these designs are typically studied and evaluated in isolation. There is no existing package that integrates these components or benchmarks them collectively. This makes it difficult to understand, select, and combine these components to effectively train LTSMs in practice.

To address the gaps, we introduce `LTSM-Bundle`, a comprehensive toolbox and benchmark for training LLMs for time series forecasting, spanning pre-processing techniques, model configurations, and dataset configurations, as depicted in Figure 1. The goal of `LTSM-Bundle` is designed to serve as a foundation for benchmarking both current and emerging LTSMs and TSFMs, while maintaining full transparency about its current model and dataset coverage. We modularize and benchmark LTSMs under the same settings and from multiple dimensions, including prompting strategies, tokenization approaches, training paradigms, base model selection, data quantity, and dataset diversity. In addition to existing components, we introduce *time series prompt*, a statistical prompting strategy tailored for time series data, as one of our benchmarking components. It generates prompts by extracting global features from the training dataset, providing robust statistical descriptions for heterogeneous data.

Through extensive benchmarking with `LTSM-Bundle`, we combine the most effective design choices identified in our study for training LTSMs. Our empirical results suggest that the identified combination produces superior zero-shot and few-shot (with 5% training data) performances compared to the state-of-the-art LTSMs in benchmark data sets. Additionally, even with just 5% of the data, our result is comparable to the strong baselines trained on the full training data, demonstrating the practical value of `LTSM-Bundle` in developing and training LTSMs. In summary, we have made the following contributions:

- We present `LTSM-Bundle`, a comprehensive toolbox and benchmark for LTSMs. `LTSM-Bundle` not only includes various components with easy-to-use interfaces for training LTSMs but is also integrated with TDengine, a state-of-the-art time series database, to build up an end-to-end training pipeline from time series data storage to report visualization and generation.

- We perform systematic analysis with `LTSM-Bundle`. Our analysis yields numerous insightful observations, paving the path for future research endeavors.
- Tokenization design critically affects cross-architecture transferability. Our results show that different tokenization strategies have a significant impact on forecasting performance across diverse model architectures.
- Model size is not always correlated with accuracy. We find that smaller and medium-sized models can achieve competitive or even superior performance compared to larger models, particularly on long-horizon forecasting tasks.

## 2 Notations and Problem Formulation

We denote a multi-variate time series as $\mathbf{Z} = \{\mathbf{z}_1, \mathbf{z}_2, ..., \mathbf{z}_T\}$, where $\mathbf{z}_t \in \mathbb{R}^d$ is a vector of multivariate variable with dimension $d$, and $T$ is the total number of timestamps. We typically partition $\mathbf{Z}$ chronologically to create training, validation, and testing sets, denoted as $\mathbf{Z}^{\text{train}} = \{\mathbf{z}_1, \mathbf{z}_2, ..., \mathbf{z}_{T^{\text{train}}}\}$, $\mathbf{Z}^{\text{val}} = \{\mathbf{z}_{T^{\text{train}}+1}, \mathbf{z}_{T^{\text{train}}+2}, ..., \mathbf{z}_{T^{\text{train}}+T^{\text{val}}}\}$, and $\mathbf{Z}^{\text{test}} = \{\mathbf{z}_{T^{\text{train}}+T^{\text{val}}+1}, \mathbf{z}_{T^{\text{train}}+T^{\text{val}}+2}, ..., \mathbf{z}_T\}$, where $T^{\text{train}}$ and $T^{\text{test}}$ denote the number of timestamps for training and validation, respectively. In traditional TSF, we aim to train a model using $\mathbf{Z}^{\text{train}}$ such that, on $\mathbf{Z}^{\text{test}}$, given the observations from the historical $P$ timestamps $\mathbf{X} = \{\mathbf{z}_{t_1}, \mathbf{z}_{t_2}, ..., \mathbf{z}_{t_P}\}$, the model can accurately predict the values of future $Q$ timestamps $\mathbf{Y} = \{\mathbf{z}_{t_{P+1}}, \mathbf{z}_{t_{P+2}}, ..., \mathbf{z}_{t_{P+Q}}\}$, where $\mathbf{X}$ and $\mathbf{Y}$ are sub-sequences of $\mathbf{Z}^{\text{test}}$. In our work, the LTSMs are trained by minimizing mean square error loss $\mathcal{L}(\text{LTSM}(\mathbf{X}), \mathbf{Y})$ between the given sub-sequences.

We focus on training LTSMs, where the objective is to develop a model that performs well across various test sets, denoted as $\mathcal{Z}^{\text{test}} = \{\mathbf{Z}_1^{\text{test}}, \mathbf{Z}_2^{\text{test}}, ..., \mathbf{Z}_N^{\text{test}}\}$, where $N$ represents the number of datasets for testing. Each $\mathbf{Z}_i^{\text{test}}$ may originate from a distinct domain, with different lengths, dimensions, and frequencies. The training sets for an LTSM may comprise training data associated with $\mathcal{Z}^{\text{test}}$ or data from other sources, provided they are not included in $\mathcal{Z}^{\text{test}}$. Training LTSMs presents a notable challenge compared to traditional TSF models due to the inherent difficulty in accommodating diverse patterns across datasets, often necessitating specialized designs. Nevertheless, it also offers opportunities to transfer knowledge from existing time series to new scenarios.

## 3 `LTSM-Bundle` Library

### 3.1 Package Design

We showcase the system overview of `LTSM-Bundle` in Figure 2. `LTSM-Bundle` is a modular toolkit that supports the complete life-cycle of large time–series models (LTSMs), from raw data ingestion to deployment–ready evaluation. The framework is organized around four interoperable subsystems, all exposed through a unified API that eliminates boilerplate engineering and accelerates experimentation. First, **TS Tokenizing** converts multivariate time series into token sequences via linear and dynamic schemes that preserve both global trends and local temporal dynamics, making the data directly consumable by Transformer-style backbones. Second, **TS Prompting** embeds task instructions and statistical context—through hard, soft, and statistics-aware prompts—into the token stream, enabling zero- and few-shot adaptation to forecasting, anomaly detection, and classification tasks. Third, the **Data Processing** layer supplies scalable loaders, windowing utilities, and feature-engineering pipelines that abstract away dataset idiosyncrasies and seamlessly handle large benchmarks. Fourth, **LTSM Training** offers a uniform optimization interface for fine-tuning or training from scratch a range of backbone architectures and parameter scales, with built-in support for curriculum learning, transfer learning, and large-scale hyperparameter sweeps. These subsystems are orchestrated by a reproducible workflow engine that chains tokenizing, prompting, processing, and training steps into end-to-end pipelines. The toolkit further provides a library of loss functions, data-augmentation routines, and evaluation metrics, alongside visualization and reporting utilities that generate publication-ready artifacts. Collectively, `LTSM-Bundle` delivers a comprehensive and extensible platform for constructing, analyzing, and deploying large time–series models, thereby lowering the barrier to reproducible research and accelerating industrial adoption.

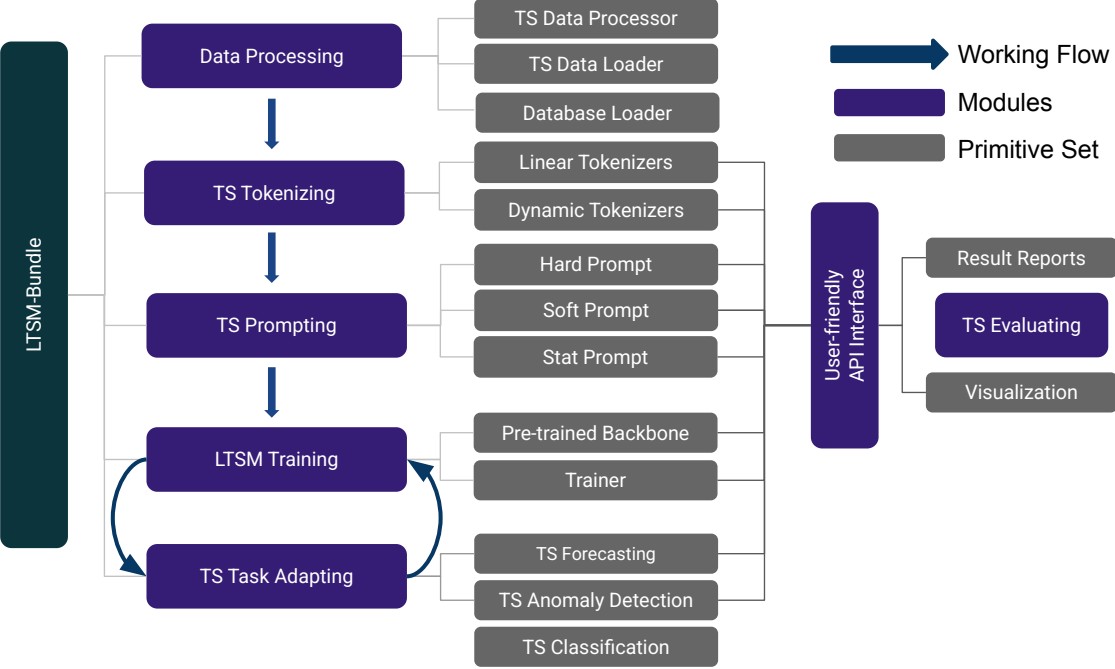

Figure 2: System Overview of LTSM-Bundle library. LTSM-Bundle provides an end-to-end training and evaluation pipeline constructed from data preprocessing to visualization. It also provides a database reader to better incorporate with large-scale datasets.

## 3.2 Package Interface

The `LTSM-Bundle` package is implemented with a scikit-learn-like interface to easily train a customized LTSM. To increase the flexibility of `LTSM-Bundle` in utilizing a broader spectrum of backbone models and training paradigms, we have integrated it with the Huggingface Transformers package[1]. This integration allows for the incorporation of diverse pre-trained weights and supports various training approaches, enhancing the overall adaptability of our framework. Researchers and practitioners only need to provide their own time series data and specify the chosen training configurations; then, every training pipeline can be created using the `LTSM-Bundle` package. Moreover, `LTSM-Bundle` supports linking with time series vector database, TDengine[2], a state-of-the-art time series database, to build an end-to-end training pipeline from time series data storage to report visualization and generation. Without struggling to build up extra efforts on clear report generation and visualization, `LTSM-Bundle` automatically generates all results with an easy-to-read template at the top of TDengine.

## 4 Benchmarking LTSM Training

We benchmark existing LTSM components by involving and coordinating them into four fundamental components of `LTSM-Bundle` package. We aim to answer the following research questions: **1) How do tokenization methods and prompting techniques impact model convergence? 2) How do base model selection and the training paradigms impact the time series forecasting performance?** and **3) How do different dataset configurations impact the model generalization?**

To answer the questions, we evaluate the impact of each component individually by keeping the rest of the components fixed. First, we fix the base model and dataset configurations with smaller model sizes, limited data quantities, and reduced data diversity, excluding prompt tokenization, to identify the optimal prompting

---

[1]https://github.com/huggingface/transformers
[2]https://github.com/taosdata/TDengine

strategy. Next, we incorporate the best prompting strategy with the same base model selection and dataset configuration to assess the tokenization methods. Afterward, we keep all the components constant except the base model to study the impact of different model initialization and training strategies. Finally, with the best tokenization and prompting methods, we select a list of candidate base models following the guidelines from the previous step. We also control the quantity and diversity of the training data to assess their impacts on model generalizability and prediction performances.

We follow the experimental settings outlined in Timesnet (Wu et al., 2022) and Time-LLM (Jin et al., 2023), employing the unified evaluation framework. The input time series length $\mathcal{E}$ is set to 336, with four different prediction lengths in $\{96, 192, 336, 720\}$. Evaluation metrics include mean square error (MSE) and mean absolute error (MAE). We also calculate the average scores among all prediction horizons. The results highlighted in **red** represent the **best** performance and highlighted in **blue** represent the **second best** performance. The details of hyperparameter settings of our banchmarking experiments are in Appendix B, respectively.

Our evaluations use widely adopted benchmarks: the ETT series (ETTh1, ETTh2, ETTm1, ETTm2)(Zhou et al., 2021), Traffic, Electricity, Weather, and Exchange-Rate datasets(Lai et al., 2018; Wu et al., 2022). ETT comprises four subsets—two with hourly data (ETTh) and two with 15-minute data (ETTm)—each containing seven features from July 2016 to July 2018. The Traffic dataset provides hourly road occupancy rates from San Francisco freeways (2015–2016); the Electricity dataset records hourly consumption for 321 clients (2012–2014); the Weather dataset offers 21 indicators every 10 minutes in Germany during 2020; and the Exchange-Rate dataset includes daily rates for eight countries (1990–2016). For more details, refer to Appendix A.

We first train our framework on the diverse time series data collection using `LTSM-Bundle` package and then assess the best combination identified on joint learning and zero-shot transfer learning to different domains of time series knowledge. For clarity, we use the term "`LTSM-Bundle`" to represent the best practice of the combination under the described experiment settings in each of the following sections.

## 4.1 Pre-processing: Instruction Prompts

The pre-processing step plays a crucial role in enabling LLM-based models to better adapt to time series datasets. In this section, we present a detailed analysis aimed at recommending the most effective pre-processing prompting strategy to compose `LTSM-bundle`. Instruction prompts enhance the effectiveness of LTSM training by providing auxiliary information. This prompt helps the model adjust its internal state and focus more on relevant features in different domains of the dataset, thereby improving learning accuracy. With the aid of prompts, LTSM aims to optimize forecasting ability across diverse dataset domains. We explore two types of prompts: the *Text Prompts* (Jin et al., 2023) written in task-specific information, and the *time series prompts* developed by global features of time series data. This comparison determines the most effective prompt type for LTSM training.

**Time Series Prompts** Time series prompts aim to capture the comprehensive properties of time series data. Unlike instruct prompts, they are derived from a diverse set of global features extracted from the entire training dataset. This approach ensures a robust representation of the underlying dynamics, in addition to enhancing model performance. The time series prompts are generated by extracting global features from each variate of the time series training data. The extracted global features are specified in Appendix D. After extracting the global features, we proceed to standardize their values across all varieties and instances within the dataset. This standardization is crucial to prevent the overflow issue during both training and inference stages. Let $\mathbf{P} = \{\boldsymbol{p}_1, \cdots, \boldsymbol{p}_M\}$ denote the global features of $\boldsymbol{Z}$ after the standardization, where $\boldsymbol{p}_t \in \mathbb{R}^d$. Subsequently, $\mathbf{P}$ serves as prompts, being concatenated with each timestamp $\mathbf{X}$ derived from the time series data. Consequently, the large time series models take the integrated vector $\tilde{\mathbf{X}} = \mathbf{P} \cup \mathbf{X} = \{\boldsymbol{p}_1, \cdots, \boldsymbol{p}_M, \mathbf{z}_{t_1}, \mathbf{z}_{t_2}, ..., \mathbf{z}_{t_P}\}$ as input data throughout both training and inference phases, as illustrated in Figure 3. The time series prompts are generated separately for the training and testing datasets, without leaking the information from testing data. We leverage the package[3] to generate the time series prompts.

---

[3] https://github.com/thuml/Time-Series-Library

Table 1: Performance of different prompting strategies

| Metric | Input | ETTh1 | ETTh2 | ETTm1 | ETTm2 | Traffic | Weather | Electricity | Avg. |
|--------|-------|-------|-------|-------|-------|---------|---------|-------------|------|
| **MSE** | No prompt | 0.308 | 0.237 | 0.367 | 0.157 | 0.306 | 0.177 | 0.148 | 0.243 |
| | TS prompt | 0.307 | 0.234 | 0.285 | 0.155 | 0.305 | 0.172 | 0.145 | 0.229 |
| | Text prompt | 0.319 | 0.241 | 0.490 | 0.190 | 0.345 | 0.212 | 0.185 | 0.283 |
| **MAE** | No prompt | 0.375 | 0.325 | 0.411 | 0.258 | 0.272 | 0.232 | 0.246 | 0.303 |
| | TS prompt | 0.377 | 0.326 | 0.369 | 0.266 | 0.279 | 0.242 | 0.247 | 0.301 |
| | Text prompt | 0.386 | 0.329 | 0.476 | 0.289 | 0.326 | 0.269 | 0.299 | 0.339 |

Table 2: Performance of linear and time series tokenization

| Metric | Tokenizer | ETTh1 | ETTh2 | ETTm1 | ETTm2 | Traffic | Weather | Exchange | Electricity | Avg. |
|--------|-----------|-------|-------|-------|-------|---------|---------|----------|-------------|------|
| **MSE** | Linear tokenizer | 0.301 | 0.228 | 0.261 | 0.149 | 0.300 | 0.163 | 0.058 | 0.140 | 0.214 |
| | Time series tokenizer | 1.798 | 0.855 | 1.671 | 0.625 | 2.199 | 0.983 | 3.729 | 2.206 | 1.663 |
| **MAE** | Linear tokenizer | 0.372 | 0.319 | 0.346 | 0.265 | 0.268 | 0.230 | 0.173 | 0.241 | 0.281 |
| | Time series tokenizer | 1.057 | 0.606 | 0.991 | 0.488 | 1.083 | 0.619 | 1.495 | 1.108 | 0.895 |

**Experimental Results** We begin by evaluating the effectiveness of instruction prompts. Specifically, we assess two distinct types of instruction prompts, both initialized by the same pre-trained GP2-Medium weights within the context of commonly used linear tokenization. The experimental results are shown in Table 1. Our observations suggest that ① statistical prompts outperform traditional text prompts in enhancing the training of LTSM models with up to 8% lower MAE scores. Additionally, ② it is observed that the use of statistical prompts results in superior performance compared to scenarios where no prompts are employed, yielding up to 3% lower MSE scores. The superiority of statistical prompt is evident in the more effective leveraging of LTSM capabilities, leading to improved learning outcomes across various datasets. Based on the above observations, we select time series prompts as the focus in the following analysis and incorporate them into `LTSM-bundle`.

## 4.2 Pre-processing: Tokenizations

In addition to employing instructional prompts to enhance generalization in LTSM training, this section provides a detailed analysis aimed at identifying the most effective tokenization strategy for LTSMs. We explore two distinct tokenization approaches – linear tokenization (Zhou et al., 2023) and time series tokenization (Ansari et al., 2024) – to determine the superior method for training LTSM models.

**Details of Tokenization** To harness the power of LLMs, a prevalent strategy involves mapping time series values to tokens (Zhou et al., 2023; Jin et al., 2023). However, converting time series data to natural language formats for LLMs is not trivial, as LLMs are pre-trained with predetermined tokenizers designed for NLP datasets. However, this implies that time series data cannot be directly fed into LLMs for training on forecasting purposes; it requires a specialized transformation of the time series data into specific indices suitable for processing by the LLMs. In this manner, we utilize two advanced types of tokenizations, linear tokenization and time series tokenization, to better evaluate their effectiveness in transferring data for training LTSMs. Specifically, the linear tokenization (Zhou et al., 2023) leverages one trainable linear layer $f : \mathbb{R}^{\mathcal{E}} \to \mathbb{R}^{K}$ to transfer time series numbers to specific tokens, where $\mathcal{E}$ denotes time series length, and $K$ refers to input size of pre-trained LLM backbone. The trainable time series tokenization (Ansari et al., 2024) aims to covert continuous time series data into discrete tokens by scaling and quantizing their values to the specific number of token bins with a given Dirichlet function.

**Experimental Results** We investigate the impact of two tokenization methods on training LTSMs. By comparing different tokenization strategies, we aim to identify which approach best complements the LTSM architecture, enhancing its ability to process and learn from complex and multi-domain datasets. Specifically, we conduct experiments comparing linear tokenization and time series tokenization, utilizing pre-trained GPT-2-medium models along with time series prompts. The experimental results shown in Table 2 demon-

Table 3: Performance of learning from scratch, LoRA fine-tuning, and fully fine-tuning

| | Metric | MSE | | | | MAE | | | |
|---|---|---|---|---|---|---|---|---|---|
| | Predict length | 96 | 192 | 336 | 720 | 96 | 192 | 336 | 720 |
| **TS prompt** | From scratch | 0.325 | 0.296 | 0.323 | 0.355 | 0.355 | 0.375 | 0.374 | 0.409 |
| | LoRA fine-tuning | 0.343 | 0.381 | 0.399 | 0.466 | 0.374 | 0.403 | 0.426 | 0.478 |
| | Fully fine-tuning | 0.229 | 0.260 | 0.297 | 0.351 | 0.301 | 0.324 | 0.354 | 0.397 |
| **Text prompt** | From scratch | 0.494 | 0.434 | 0.597 | 0.485 | 0.463 | 0.438 | 0.512 | 0.475 |
| | LoRA fine-tuning | 0.347 | 0.379 | 0.406 | 0.473 | 0.373 | 0.404 | 0.431 | 0.484 |
| | Fully fine-tuning | 0.294 | 0.286 | 0.353 | 0.358 | 0.330 | 0.338 | 0.378 | 0.429 |

strate that linear tokenization more effectively facilitates the training process of LTSM compared to time series tokenization. In our study, we focus on smaller and more accessible training data. Under these circumstances, we observe that a linear tokenization is a more suitable choice than a time series tokenization, as the pre-trained time series tokenization may not have the transferability toward different model architecture. Unlike textual tokenization, the time series tokenization is determined by a pre-trained time series dataset with a designated LTSM architecture In summary, ③ linear tokenization is flexible and adaptive for different settings of LTSM training compared to time series tokenization under a smaller amount of training data.

### 4.3 Model Configuration: Training Paradigm

Different training paradigms exhibit unique characteristics that influence how well LLMs fit a specific training dataset. In this section, we explore three distinct training paradigms, fully fine-tuning, training from scratch, and LoRA (Hu et al., 2021), to identify the most effective approaches for training the LTSM framework.

**Training Paradigm.** In the full fine-tuning paradigm, we utilize the pre-trained weights of each base LLM, which finetune all parameters using the given time series dataset. Conversely, in the training-from-scratch paradigm, we only preserve the original model architecture but initialize all parameters anew before training with the time series dataset. In the LoRA paradigm, we employ low-rank adapters on base LLMs.

**Experimental Results.** We assess the effectiveness of the training paradigm under the settings of time series prompt and text prompt usage. Table 3 presents the results of various training strategies using GPT-2-Medium as the backbone. In general, the experimental results indicate that full fine-tuning is the most effective strategy for training the LTSM framework whether leveraging time series prompts or text prompts. Based on the results, we summarize the observations as follows. ④ Although training-from-scratch achieves competitive performance compared to full fine-tuning, the large number of trainable model parameters may lead to overfitting, ultimately degrading performance. ⑤ Fully fine-tuning paradigm leads to the best performance with up to 11% of improvement on MSE and up to 17% of improvement on MAE under the length of {96, 192, 336}, and performance competitive under the length of 720. Training `LTSM-bundle` under the full fine-tuning paradigm is recommended, as it converges twice as fast as training from scratch, ensuring efficient and effective forecasting.

### 4.4 Model Configuration: Base Model Selection

**Base Model Candidates** As for the base models of our framework, we leverage four different pre-trained models, including GPT-2-small, GPT-2-medium, GPT-2-large (Radford et al., 2019), and Phi-2 (Javaheripi & Bubeck, 2023). GPT-2 employs a transformer architecture with up to 48 layers, and it is trained on a diverse corpus of internet text, resulting in a model size of 124M (small), 355M (medium), and 774M (large) parameters. Phi-2 also uses a transformer-based architecture but emphasizes high-quality ("textbook-quality") data, comprising 2.7 billion parameters. Despite its smaller size compared to the largest contemporary models, Phi-2 incorporates innovative scaling techniques to optimize performance. Different from the absolute positional encoding used by GPT-2, Phi-2 employs relative positional encoding, which considers the pairwise distance between each token pair for encoding position information of tokens. Following the settings

Table 4: Performance of different backbones

| Metric | MSE | | | | MAE | | | |
|--------|-----|-----|-----|-----|-----|-----|-----|-----|
| **GPT-2** | **96** | **192** | **336** | **720** | **96** | **192** | **336** | **720** |
| Small | 0.252 | 0.306 | 0.316 | 0.352 | 0.313 | 0.363 | 0.367 | 0.400 |
| Medium | 0.229 | 0.260 | 0.297 | 0.351 | 0.301 | 0.324 | 0.354 | 0.397 |
| Large | 0.224 | 0.257 | 0.301 | 0.358 | 0.292 | 0.322 | 0.356 | 0.399 |

Table 5: Performance of different down-sampling rates

| Metric | MSE | | | | MAE | | | |
|--------|-----|-----|-----|-----|-----|-----|-----|-----|
| **DS rate** | **96** | **192** | **336** | **720** | **96** | **192** | **336** | **720** |
| 2.5% | 0.227 | 0.268 | 0.308 | 0.369 | 0.294 | 0.335 | 0.365 | 0.415 |
| 5% | 0.229 | 0.261 | 0.297 | 0.350 | 0.301 | 0.324 | 0.354 | 0.397 |
| 10% | 0.233 | 0.260 | 0.291 | 0.350 | 0.289 | 0.323 | 0.348 | 0.396 |

in (Zhou et al., 2023), we utilize the top three self-attention layers of every pre-trained model as our backbone structure in `LTSM-bundle` framework.

**Experimental Results** We explore the impact of using different pre-trained LLM weights as backbones in LTSM models, with the goal of identifying the most suitable pre-trained LLM weights for processing time series data. The findings are detailed in Table 4. We assess the performance of different backbones with time series prompts under the fully fine-tuning paradigm. We summarize our observations as follows: ⑥ GPT-2-Small demonstrates a performance improvement of up to 2% in relatively long-term forecasting (i.e., 336 and 720 hours) compared to the GPT-2-Large model. ⑦ GPT-2-Medium outperforms GPT-2-Large in relatively short-term forecasting (i.e., 96 and 192 hours), as larger models may be prone to overfitting during training, degrading forecasting performance.

While our benchmarking results in Table 4 indicate that variations in the number of parameters within the same architecture have minimal impact on performance, we did not limit our analysis to a single model size alone. To ensure the rigor of our evaluation, we compared Phi-2—an alternative model architecture—with GPT-2 models of varying sizes (large, medium, and small) using different prompting methods, as detailed in Table 18 and Table 19 of Appendix F. The results show that GPT-2 Small and Medium obtains higher performance than Phi-2 on both time series prompt and textual instruction prompt settings. Based on the above findings, we recommend incorporating GPT-2-Medium or GPT-2-Small as the backbone of `LTSM-bundle`.

### 4.5 Dataset configuration: Quantity

The quantity of datasets is often the key to the success of LLMs due to the consistent semantic meaning of tokens. Nevertheless, time series tokens are less informative and semantically meaningful compared to natural language tokens. In this section, we investigate the impact of data quantity to determine whether the principle that more training data leads to better LTSMs.

**Quantity Configuration** We conduct the down-sampling strategies to study the impact of data quantity on prediction performance. Specifically, each time series in the training data are periodically down-sampled along the timestamps to reduce the granularity of the entire time series while maintaining the general pattern. Each dataset is split into training, validation, and testing sets, and then down-sampling is applied to the training set for model training. We compare the models trained with 10%, 5%, and 2.5% of the full-size time series in the training set. In the following experiment section, we annotate partial training data usage as few-shot training.

**Experimental Results** Table 5 lists the results of models trained with different data quantities under GPT-2 Medium as the model backbone. The model trained with 5% and 10% down-sampled data leads to the best result compared to 2.5%. ⑧ We observe that increasing the amount of data does not positively correlate with improved model performance. The rationale is that increasing data points enhances the granularity of

Table 6: Average performance with different downsampling under different domain `LTSM-Bundle`

| (MAE/MSE) | 1 dataset | 2 datasets | 4 datasets | 8 datasets |
|---|---|---|---|---|
| 2.5% | 0.446/0.450 | **0.435/0.485** | 0.396/0.357 | 0.352/0.293 |
| 5% | 0.416/0.380 | **0.415/0.436** | 0.383/0.341 | 0.344/0.283 |
| 10% | 0.414/0.375 | **0.415/0.440** | 0.394/0.355 | 0.348/0.288 |

Table 7: Performance of LTSM trained on different numbers of datasets

|  |  | 1 dataset | 2 datasets | 3 datasets | 4 datasets | 5 datasets | 6 datasets | 7 datasets | 8 datasets |
|---|---|---|---|---|---|---|---|---|---|
| **MSE** | 96 | 0.333 | 0.366 | 0.269 | 0.276 | 0.232 | 0.229 | 0.227 | 0.215 |
|  | 192 | 0.394 | 0.440 | 0.309 | 0.318 | 0.271 | 0.267 | 0.269 | 0.254 |
|  | 336 | 0.403 | 0.427 | 0.356 | 0.351 | 0.302 | 0.325 | 0.308 | 0.302 |
|  | 720 | 0.478 | 0.511 | 0.436 | 0.419 | 0.373 | 0.364 | 0.369 | 0.361 |
| **MAE** | 96 | 0.351 | 0.351 | 0.327 | 0.329 | 0.298 | 0.294 | 0.292 | 0.282 |
|  | 192 | 0.407 | 0.395 | 0.363 | 0.368 | 0.333 | 0.332 | 0.334 | 0.323 |
|  | 336 | 0.420 | 0.420 | 0.401 | 0.392 | 0.361 | 0.384 | 0.371 | 0.362 |
|  | 720 | 0.464 | 0.494 | 0.466 | 0.445 | 0.423 | 0.411 | 0.419 | 0.411 |

time series but may reduce the model's generalization ability, while excessive down-sampling loses critical information, hindering pattern learning. Thus, optimizing model performance requires carefully balancing the amount of training data with its diversity. To explore this balance, we conducted experiments using different downsampling rates across various time series datasets. After comparing with the different downsampling rates (i.e., 2.5%, 10%, and 25% in Table 5 and Table 6). We use 5% of the dataset to benchmark the LTSM-Bundle family across diverse time series data, as increasing the dataset size beyond this point results in only marginal performance improvements while significantly increasing training time (i.e., 10% increase doubles the computational cost but yields only a slight enhancement in forecasting performance).

### 4.6 Dataset Configuration: Diversity

**Impact of Dataset Diversity** Recall that we utilized eight datasets for training purposes, encompassing ETT variant, Traffic, Electricity, Weather, and Exchange. Here, we focus on evaluating the performance of LTSM models when trained with subsets of these datasets. Specifically, we employ the first $M$ datasets from the aforementioned list for training, where $M \in 1, 2, ..., 8$. For instance, when $M = 1$, solely ETTh1 is utilized for training; when $M = 5$, ETTh1, ETTh2, ETTm1, ETTm2, and Weather are utilized. Subsequently, we evaluate the trained model's performance across all datasets to understand the impact of dataset diversity.

**Experimental Results** Table 7 summarizes the results. ⑨ Augmenting dataset diversity generally leads to improved performance. This is expected because more diverse data has the potential to enhance the generalization capabilities of LTSMs across various patterns. We conclude two reasons: (1) Although originating from different domains with distinct characteristics, datasets may share underlying knowledge that can be transferred, enhancing model generalization and performance. (2) Prompting strategies, particularly time series prompts, can further facilitate knowledge transfer by guiding the model to implicitly learn which information to retain and which to discard.

## 5 Comparison with Baselines

Based on the observations in Section 4, we identify a strong combination using `LTSM-Bundle` with the settings as follows: (1) Base model backbone: GPT-2-Medium, (2) Instruction prompts: the time series prompts, (3) Tokenization: linear tokenization, and (4) Training paradigm: fully fine-tuning. We compare this combination against SoTA TSF models on zero-shot and few-shot settings.

Table 8: Zero-shot performance. "`LTSM-Bundle`" denotes the best combination of LTSM training components

| Metric | LTSM-Bundle | | TIME-LLM | | GPT4TS | | LLMTime | | DLinear | | PatchTST | | TimesNet | |
|---|---|---|---|---|---|---|---|---|---|---|---|---|---|---|
| | MSE | MAE | MSE | MAE | MSE | MAE | MSE | MAE | MSE | MAE | MSE | MAE | MSE | MAE |
| ETTh1 → ETTh2 | 0.319 | 0.402 | 0.353 | 0.387 | 0.406 | 0.422 | 0.992 | 0.708 | 0.493 | 0.488 | 0.380 | 0.405 | 0.421 | 0.431 |
| ETTh1 → ETTm2 | 0.312 | 0.406 | 0.273 | 0.340 | 0.325 | 0.363 | 1.867 | 0.869 | 0.415 | 0.452 | 0.314 | 0.360 | 0.327 | 0.361 |
| ETTm1 → ETTh2 | 0.306 | 0.391 | 0.381 | 0.412 | 0.433 | 0.439 | 0.992 | 0.708 | 0.464 | 0.475 | 0.439 | 0.438 | 0.457 | 0.454 |
| ETTm1 → ETTm2 | 0.217 | 0.319 | 0.268 | 0.320 | 0.313 | 0.348 | 1.867 | 0.869 | 0.335 | 0.389 | 0.296 | 0.334 | 0.322 | 0.354 |
| ETTm2 → ETTh2 | 0.314 | 0.393 | 0.354 | 0.400 | 0.435 | 0.443 | 1.867 | 0.869 | 0.455 | 0.471 | 0.409 | 0.425 | 0.435 | 0.443 |
| ETTm2 → ETTm1 | 0.403 | 0.430 | 0.414 | 0.438 | 0.769 | 0.567 | 1.933 | 0.984 | 0.649 | 0.537 | 0.568 | 0.492 | 0.769 | 0.567 |

## 5.1 Experimental Settings

We follow the same settings as in Time-LLM (Jin et al., 2023). Specifically, for zero-shot experiments, we test the model's cross-domain adaptation under the long-term forecasting scenario and evaluate it on various cross-domain scenarios utilizing the ETT datasets. The hyperparameter settings of training `LTSM-Bundle` are in Appendix B. For the few-shot setting, we train our `LTSM-Bundle` on 5% of the data and compare it with other baselines under the 5% as well. We cite the performance of other models when applicable (Zhou et al., 2023). Furthermore, we compare `LTSM-Bundle` trained on 5% training data against baselines trained on the full training set. Our findings in Appendix F indicate that `LTSM-Bundle` achieves comparable results, further underscoring its superiority.

The baseline methods consist of various Transformer-based methods, including PatchTST (Nie et al., 2022), ETSformer (Woo et al., 2022), Non-Stationary Transformer (Liu et al., 2022), FEDformer (Zhou et al., 2022), Autoformer (Chen et al., 2021), Informer (Zhou et al., 2021), and Reformer (Kitaev et al., 2020). Additionally, we evaluate our model against recent competitive models like Time-LLM (Jin et al., 2023), TEST (Sun et al., 2023), LLM4TS (Chang et al., 2023), GPT4TS (Zhou et al., 2023), DLinear (Zeng et al., 2023), TimesNet (Wu et al., 2022), and LightTS (Campos et al., 2023). More details of the baseline methods can be found in Section 6.

## 5.2 Zero-shot and Few-shot Results

**Zero-shot Performance** In the zero-shot learning experiments shown in Table 8 shows that the best component combination from benchmarking `LTSM-Bundle` consistently delivers superior performance across various cross-domain scenarios using the ETT datasets. For example, in the ETTh1 to ETTh2 dataset transfer task, `LTSM-Bundle` achieves an MSE of 0.319 and an MAE of 0.402, outperforming all other methods, including TIME-LLM, GPT4TS, and DLinear. Similarly, in the ETTm1 to ETTm2 dataset transfer scenario, `LTSM-Bundle` records the lowest MSE and MAE scores of 0.217 and 0.319, showing its strong generalization capability across different domains. The consistent improvements across transfer tasks of `LTSM-Bundle` in zero-shot learning.

**Few-shot Performance** Table 9 presents the performance of the best component combination from benchmarking compared to the baseline models in the few-shot setting, utilizing 5% of the training data. Notably, `LTSM-Bundle` exhibits a significant advantage over both traditional baselines and existing LTSMs. Across the 7 datasets, `LTSM-Bundle` outperforms all baselines regarding MSE in 5 datasets and regarding MAE in 4 datasets. Moreover, `LTSM-Bundle` achieves the top rank 40 times among the reported results. These findings underscore the effectiveness of our model in few-shot scenarios, where it demonstrates high accuracy even with limited training data. Its capability to excel with minimal data not only highlights its adaptability but also its potential for practical applications, particularly in contexts where data availability is constrained. All further and full versions of results on the full datasets are provided in Appendix F.

## 6 Related Works

Table 9: Performance comparison in the few-shot setting with 5% training data. The full results are provided in Appendix F. "`LTSM-Bundle`" denotes the best combination of LTSM training components

| | Metric | LTSM-Bundle | | TIME-LLM | | LLM4TS | | GPT4TS | | DLinear | | PatchTST | | TimesNet | | FEDformer | |
|---|---|---|---|---|---|---|---|---|---|---|---|---|---|---|---|---|---|
| | | MSE | MAE | MSE | MAE | MSE | MAE | MSE | MAE | MSE | MAE | MSE | MAE | MSE | MAE | MSE | MAE |
| ETTh1 | 96 | 0.307 | 0.377 | 0.483 | 0.464 | 0.509 | 0.484 | 0.543 | 0.506 | 0.547 | 0.503 | 0.557 | 0.519 | 0.892 | 0.625 | 0.593 | 0.529 |
| | 192 | 0.329 | 0.391 | 0.629 | 0.540 | 0.717 | 0.581 | 0.748 | 0.580 | 0.720 | 0.604 | 0.711 | 0.570 | 0.940 | 0.665 | 0.652 | 0.563 |
| | 336 | 0.346 | 0.405 | 0.768 | 0.626 | 0.728 | 0.589 | 0.754 | 0.595 | 0.984 | 0.727 | 0.816 | 0.619 | 0.945 | 0.653 | 0.731 | 0.594 |
| | 720 | 0.370 | 0.441 | - | - | - | - | - | - | - | - | - | - | - | - | - | - |
| | Avg | 0.338 | 0.403 | 0.627 | 0.543 | 0.651 | 0.551 | 0.681 | 0.560 | 0.750 | 0.611 | 0.694 | 0.569 | 0.925 | 0.647 | 0.658 | 0.562 |
| ETTh2 | 96 | 0.235 | 0.326 | 0.336 | 0.397 | 0.314 | 0.375 | 0.376 | 0.421 | 0.442 | 0.456 | 0.401 | 0.421 | 0.409 | 0.420 | 0.390 | 0.424 |
| | 192 | 0.283 | 0.365 | 0.406 | 0.425 | 0.365 | 0.408 | 0.418 | 0.441 | 0.617 | 0.542 | 0.452 | 0.455 | 0.483 | 0.464 | 0.457 | 0.465 |
| | 336 | 0.320 | 0.401 | 0.405 | 0.432 | 0.398 | 0.432 | 0.408 | 0.439 | 1.424 | 0.849 | 0.464 | 0.469 | 0.499 | 0.479 | 0.477 | 0.483 |
| | 720 | 0.378 | 0.456 | - | - | - | - | - | - | - | - | - | - | - | - | - | - |
| | Avg | 0.303 | 0.387 | 0.382 | 0.418 | 0.359 | 0.405 | 0.400 | 0.433 | 0.694 | 0.577 | 0.439 | 0.448 | 0.439 | 0.448 | 0.463 | 0.454 |
| ETTm1 | 96 | 0.285 | 0.369 | 0.316 | 0.377 | 0.349 | 0.379 | 0.386 | 0.405 | 0.332 | 0.374 | 0.399 | 0.414 | 0.606 | 0.518 | 0.628 | 0.544 |
| | 192 | 0.319 | 0.393 | 0.450 | 0.464 | 0.374 | 0.394 | 0.440 | 0.438 | 0.358 | 0.390 | 0.441 | 0.436 | 0.681 | 0.539 | 0.666 | 0.566 |
| | 336 | 0.378 | 0.425 | 0.450 | 0.424 | 0.411 | 0.417 | 0.485 | 0.459 | 0.402 | 0.416 | 0.499 | 0.467 | 0.786 | 0.597 | 0.807 | 0.628 |
| | 720 | 0.464 | 0.477 | 0.483 | 0.471 | 0.516 | 0.479 | 0.577 | 0.499 | 0.511 | 0.489 | 0.767 | 0.587 | 0.796 | 0.593 | 0.822 | 0.633 |
| | Avg | 0.362 | 0.416 | 0.425 | 0.434 | 0.412 | 0.417 | 0.472 | 0.450 | 0.400 | 0.417 | 0.526 | 0.476 | 0.717 | 0.561 | 0.730 | 0.592 |
| ETTm2 | 96 | 0.156 | 0.266 | 0.174 | 0.261 | 0.192 | 0.273 | 0.199 | 0.280 | 0.236 | 0.326 | 0.206 | 0.288 | 0.220 | 0.299 | 0.229 | 0.320 |
| | 192 | 0.203 | 0.307 | 0.215 | 0.287 | 0.249 | 0.309 | 0.256 | 0.316 | 0.306 | 0.373 | 0.264 | 0.324 | 0.311 | 0.361 | 0.394 | 0.361 |
| | 336 | 0.255 | 0.349 | 0.273 | 0.330 | 0.301 | 0.342 | 0.318 | 0.353 | 0.380 | 0.423 | 0.334 | 0.367 | 0.338 | 0.366 | 0.378 | 0.427 |
| | 720 | 0.342 | 0.417 | 0.433 | 0.412 | 0.402 | 0.405 | 0.460 | 0.436 | 0.674 | 0.583 | 0.454 | 0.432 | 0.509 | 0.465 | 0.523 | 0.510 |
| | Avg | 0.239 | 0.335 | 0.274 | 0.323 | 0.286 | 0.332 | 0.308 | 0.346 | 0.399 | 0.426 | 0.314 | 0.352 | 0.344 | 0.372 | 0.381 | 0.404 |
| Weather | 96 | 0.172 | 0.242 | 0.172 | 0.263 | 0.173 | 0.227 | 0.175 | 0.230 | 0.184 | 0.242 | 0.171 | 0.224 | 0.207 | 0.253 | 0.229 | 0.309 |
| | 192 | 0.218 | 0.278 | 0.224 | 0.271 | 0.218 | 0.265 | 0.227 | 0.276 | 0.228 | 0.283 | 0.230 | 0.277 | 0.272 | 0.307 | 0.265 | 0.317 |
| | 336 | 0.276 | 0.329 | 0.282 | 0.321 | 0.276 | 0.310 | 0.286 | 0.322 | 0.279 | 0.322 | 0.294 | 0.326 | 0.313 | 0.328 | 0.353 | 0.392 |
| | 720 | 0.339 | 0.373 | 0.366 | 0.381 | 0.355 | 0.366 | 0.366 | 0.379 | 0.364 | 0.388 | 0.384 | 0.387 | 0.400 | 0.385 | 0.391 | 0.394 |
| | Avg | 0.251 | 0.305 | 0.260 | 0.309 | 0.251 | 0.292 | 0.263 | 0.301 | 0.263 | 0.308 | 0.269 | 0.303 | 0.298 | 0.318 | 0.309 | 0.353 |
| Electricity | 96 | 0.145 | 0.247 | 0.147 | 0.242 | 0.139 | 0.235 | 0.143 | 0.241 | 0.150 | 0.251 | 0.145 | 0.244 | 0.315 | 0.389 | 0.235 | 0.322 |
| | 192 | 0.159 | 0.259 | 0.158 | 0.241 | 0.155 | 0.249 | 0.159 | 0.255 | 0.163 | 0.263 | 0.163 | 0.260 | 0.318 | 0.396 | 0.247 | 0.341 |
| | 336 | 0.180 | 0.284 | 0.178 | 0.277 | 0.174 | 0.269 | 0.179 | 0.274 | 0.175 | 0.278 | 0.183 | 0.281 | 0.340 | 0.415 | 0.267 | 0.356 |
| | 720 | 0.215 | 0.317 | 0.224 | 0.312 | 0.222 | 0.310 | 0.233 | 0.323 | 0.219 | 0.311 | 0.233 | 0.323 | 0.635 | 0.613 | 0.318 | 0.394 |
| | Avg | 0.175 | 0.276 | 0.179 | 0.268 | 0.173 | 0.266 | 0.178 | 0.273 | 0.176 | 0.275 | 0.181 | 0.277 | 0.402 | 0.453 | 0.266 | 0.353 |
| Traffic | 96 | 0.305 | 0.279 | 0.414 | 0.291 | 0.401 | 0.285 | 0.419 | 0.298 | 0.427 | 0.304 | 0.404 | 0.286 | 0.854 | 0.492 | 0.670 | 0.421 |
| | 192 | 0.313 | 0.274 | 0.419 | 0.291 | 0.418 | 0.293 | 0.434 | 0.305 | 0.447 | 0.315 | 0.412 | 0.294 | 0.894 | 0.517 | 0.653 | 0.405 |
| | 336 | 0.326 | 0.287 | 0.437 | 0.314 | 0.436 | 0.308 | 0.449 | 0.313 | 0.478 | 0.333 | 0.439 | 0.310 | 0.853 | 0.471 | 0.707 | 0.445 |
| | 720 | 0.346 | 0.301 | - | - | - | - | - | - | - | - | - | - | - | - | - | - |
| | Avg | 0.323 | 0.285 | 0.423 | 0.298 | 0.418 | 0.295 | 0.434 | 0.305 | 0.450 | 0.317 | 0.418 | 0.296 | 0.867 | 0.493 | 0.676 | 0.423 |
| $1^{st}$ Count | | 47 | | 6 | | 16 | | 0 | | 2 | | 2 | | 0 | | 0 | |

In this work, we focus on benchmarking the training paradigms of LTSMs on top of decoder-only single models. To provide a comprehensive comparison, we categorize existing LLM-based time series forecasting approaches into three representative groups: **1. Pretrained LLM Adaptation.** These methods leverage general-purpose large language models (LLMs), such as GPT, LLaMA, and Phi, for time series forecasting through fine-tuning, in-context learning, or lightweight adapter modules. Representative works include GPT4TS (Zhou et al., 2023), which adapts the Frozen Pretrained Transformer (FPT) to predict future sequences, and LLM4TS (Chang et al., 2023), which extends general-purpose LLMs for temporal modeling. **2. Time Series-Specific LLMs.** These models are explicitly designed and trained on large-scale temporal datasets to better capture temporal dependencies and improve forecasting robustness. Examples include Time-LLM (Jin et al., 2023), which treats time series data as sequential events for enhanced modeling; TEST (Sun et al., 2023), which introduces transformer enhancements for complex temporal dependencies; and other recent works such as Chronos and Moirai, which aim to establish foundation-style LLMs for time series. **3. Hybrid Architectures.** Hybrid approaches combine the strengths of LLMs with specialized temporal modeling components to enhance efficiency, interpretability, and robustness. For instance, DLinear (Zeng et al., 2023) integrates linear trend modeling with lightweight forecasting modules, while TimesNet (Wu et al., 2022) captures multiscale temporal patterns using neural operators. Similarly, LightTS (Campos et al., 2023) emphasizes efficient tokenization and fast inference for real-time applications. In addition, we benchmark widely adopted transformer-based models, including PatchTST (Nie et al., 2022), ETSformer (Woo et al., 2022), Non-Stationary Transformer (Liu et al., 2022), FEDformer (Zhou et al., 2022),

Autoformer (Chen et al., 2021), Informer (Zhou et al., 2021), and Reformer (Kitaev et al., 2020), which serve as essential baselines in our evaluation.

To the best of our knowledge, no prior art has provided a comprehensive benchmark to analyze the effectiveness of each component in the training of LTSMs. Some works (Li et al., 2024; Liu et al., 2025) maintain a fair platform to compare different time series forecasting methods. The others (Fons et al., 2024; Qiu et al., 2024) analyze the forecasting performance from the perspectives of time series patterns. This benchmark provides an accessible and modular pipeline for evaluating a diverse set of training components in LTSM development, leveraging a time series database and user-friendly visualization. With the debates on whether LLMs can benefit from time series forecasting tasks, our toolbox offers a scikit-learn-like API interface to efficiently explore each component's effectiveness in training LTSMs.

## 7 Conclusion and Future Perspectives

In this study, we present the first comprehensive toolbox and benchmark for understanding different design choices in training LLMs for time series forecasting. Our benchmark covers various aspects, including data preprocessing, model configuration, and dataset configuration. We delve into detailed design choices such as prompting, tokenization, training paradigms, base model selection, data quantity, and dataset diversity. Through this analysis, we derive 9 observations and identify the best combination for training LTSMs. We demonstrate that this combination achieves strong zero-shot and few-shot performance compared to state-of-the-art LTSMs, and it requires only 5% of the data to achieve comparable performance to state-of-the-art baselines on benchmark datasets. We hope that our findings will inspire future research in this direction, and this combination based on `LTSM-bundle` could serve as a simple yet strong baseline for future comparison.

An ongoing debate exists about whether using pre-trained weights from large language models (LLMs) can enhance time series forecasting performance (Tan et al., 2024). In response, the `LTSM-Bundle` package provides a robust platform for further investigation. Our current findings indicate that incorporating pre-trained weights can indeed improve forecasting performance. However, as LTSM training techniques evolve, future enhancements may offer additional insights and even different perspectives on the benefits of these pre-trained weights. Notably, using pre-trained weights as a starting point may reduce training time and help the models converge faster. Drawing from our analysis, we suggest two potential directions for enhancing the LTSM training components, as outlined below:

**Advancing Prompting Strategies.** The heterogeneity of time series datasets presents a significant challenge in training a universal model that can effectively fit all datasets while generalizing to unseen ones. Our analysis highlights the potential of prompting as a solution by enriching datasets with additional contextual information. Specifically, we demonstrate the effectiveness of the time series prompt, which extracts statistical insights to improve model performance. Looking ahead, we anticipate the development of more sophisticated prompting strategies to further enhance generalization. For instance, incorporating variate-specific prompts in multivariate time series data could provide richer context and improve predictive accuracy. We believe this direction holds substantial promise for future research advancements.

**Constructing Synthetic Training Data.** Our analysis highlights the significance of dataset diversity in training transferable LTSMs. Specifically, increasing the number of datasets can enhance performance significantly (observation ⑨). This insight suggests that LTSMs could achieve better transferability when exposed to more patterns during training. Thus, there is a potential for enhancing LTSMs through synthetic datasets that simulate various patterns. However, this requires more research endeavor, as synthetic data may introduce artifacts into the training dataset. This differs from previous pre-trained LTSM work like Chronos (Ansari et al., 2024), which demonstrates strong performance due to the extremely large and synthetic dataset they prepared. One of our future research directions is to augment the training data with more precise but feasible amounts of synthetic data for training LTSM for time series forecasting tasks.

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

# Appendix

## A    Details of Datasets

In this paper, the training datasets include ETT (Electricity Transformer Temperature) (Zhou et al., 2021)[4], Traffic[5], Electricity[6], Weather[7], and Exchange-Rate(Lai et al., 2018). ETT[8] (Zhou et al., 2021) comprises four subsets: two with hourly-level data (ETTh) and two with 15-minute-level data (ETTm). Each subset includes seven features related to oil and load metrics of electricity transformers, covering the period from July 2016 to July 2018. The traffic dataset includes hourly road occupancy rates from sensors on San Francisco freeways, covering the period from 2015 to 2016. The electricity dataset contains hourly electricity consumption data for 321 clients, spanning from 2012 to 2014. The weather data set comprises 21 weather indicators, such as air temperature and humidity, recorded every 10 minutes throughout 2020 in Germany. Exchange-Rate(Lai et al., 2018) contains daily exchange rates for eight countries, spanning from 1990 to 2016. We first train our framework on the diverse time series data collection and then assess the abilities of `LTSM-Bundle` on jointly learning and zero-shot transfer learning to different domains of time series knowledge.

## B    Hyper-parameter Settings of Experiments

The hyper-parameter settings of `LTSM-Bundle` training for all experiments are shown in Table 10. Other training hyper-parameters follow the default values in the `TrainingArguments` class[9] of the huggingface transformers package.

Table 10: Hyperparameter settings of `LTSM-Bundle` training

| Hyperparameter name | Value |
|---|---|
| Number of Transformer layers $N$ | 3 |
| Training / evaluation / testing split | 0.7 / 0.1 / 0.2 |
| Gradient accumulation steps | 64 |
| Learning rate | 0.001 |
| Optimizer | Adam |
| LR scheduler | CosineAnnealingLR |
| Number of epochs | 10 |
| Number of time steps per token | 16 |
| Stride of time steps per token | 8 |
| Dimensions of TS prompt | 133 |
| Transformer architectures | GPT-2-{small, medium, large}, Phi-2 |
| Length of prediction | 96, 192, 336, 720 |
| Length of input TS data | 336 |
| Data type | `torch.bfloat16` |
| Downsampling rate of training data | 20 |

## C    Computation Infrastructure

All experiments described in this paper are conducted using a well-defined physical computing infrastructure, the specifics of which are outlined in Table 11. This infrastructure is essential for ensuring the reproducibility and reliability of our results, as it details the exact hardware environments used during the testing phases.

---

[4]`https://github.com/zhouhaoyi/ETDataset`

[5]`http://pems.dot.ca.gov`

[6]`https://archive.ics.uci.edu/dataset/321/electricityloaddiagrams20112014`

[7]`https://www.bgc-jena.mpg.de/wetter/`

[8]`https://github.com/laiguokun/multivariate-time-series-data`

[9]`https://github.com/huggingface/transformers/blob/main/src/transformers/training_args.py`

Table 11: Computing infrastructure for the experiments

| Device attribute | Value |
| --- | --- |
| Computing infrastructure | GPU |
| GPU model | Nvidia A5000 / Nvidia A100 |
| GPU number | 8 × A5000 / 4 × A100 |
| GPU memory | 8 × 24GB / 4 × 80GB |

## D  Global Features for Prompts

Time series prompts are developed to encapsulate the comprehensive characteristics of time series data. Specifically, let $s$ be a certain variate of the time-series data $Z$. We identified in Table 12 the partial global features that we leverage to craft these prompts, each selected for its ability to convey critical information about the data's temporal structure and variability. For the inter-quartile and histogram in Table 12, $Q_3(s)$ and $Q_1(s)$ represent the first and third quartile of the Time series data, respectively; and $m_i$ represents the histogram in which $n$ is the total number of observations and $k$ the total number of bins.

Beyond those shown in Table 12, we also consider the global features according to the following references: Fast Fourier Transform, Wavelet transform, Zero crossing rate, Maximum peaks, Minimum peaks, ECDF percentile count, Slope, ECDF slope, Spectral distance, Fundamental frequency, Maximum frequency, Median frequency, Spectral maximum peaks (Barandas et al., 2020); Maximum Power Spectrum (Welch, 1967), Spectral Centroid (Peeters et al., 2011), Decrease (Peeters et al., 2011), Kurtosis (Peeters et al., 2011), Skewness (Peeters et al., 2011), Spread (Peeters et al., 2011), Slope (Peeters et al., 2011), Variation (Peeters et al., 2011), Spectral Roll-off (Figueira et al., 2016), Roll-on (Figueira et al., 2016), Human Range Energy (Fernandes et al., 2020), MFCC (Davis & Mermelstein, 1980), LPCC (Atal, 1974), Power Bandwidth (U., 2003), Spectral Entropy (Pan et al., 2009), Wavelet Entropy (Yan et al., 2006) and Wavelet Energy (Kocaman & Özdemir, 2009), Kurtosis (Zwillinger & Kokoska, 1999), Skewness (Zwillinger & Kokoska, 1999), Maximum (Oliphant et al., 2006), Minimum (Oliphant et al., 2006), Mean (Oliphant et al., 2006), Median (Oliphant et al., 2006) and ECDF (Raschka, 2018), ECDF Percentile (Raschka, 2018). For the implementation, we leverage the TSFEL library[10] (Barandas et al., 2020) to estimate the global features. The features are extracted separately for each variate in the time-series data.

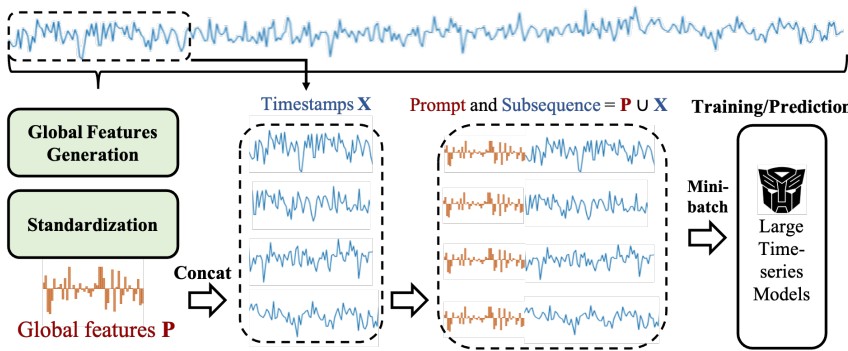

Figure 3:  Time series prompt generation process based on the given training time series data.

---

[10] https://tsfel.readthedocs.io/en/latest/descriptions/feature_list.html

Table 12: Partial global features in time series prompts

| Feature | Formula | Feature | Formula |
|---|---|---|---|
| Autocorrelation | $\sum_{i\in\mathbb{Z}} s_i s_{i-l}$ | Centroid | $\sum_{i=0}^{T} t_i \cdot s_i^2 / \sum_{i=0}^{T} s_i^2$ |
| Max differences | $\max_i(s_{i+1} - s_i)$ | Mean differences | $\text{mean}_i(s_{i+1} - s_i)$ |
| Median differences | $\text{median}_i(s_{i+1} - s_i)$ | Max absolute differences | $\max_i|s_{i+1} - s_i|$ |
| Mean absolute differences | $\text{mean}_i|s_{i+1} - s_i|$ | Median absolute differences | $\text{median}_i|s_{i+1} - s_i|$ |
| Distance | $\sum_{i=0}^{T-1} \sqrt{1 + (s_{i+1} - s_i)^2}$ | Sum of absolute differences | $\sum_{i=0}^{T-1} |s_{i+1} - s_i|$ |
| Total energy | $\sum_{i=0}^{T} s_i^2 \cdot (t_T - t_0)$ | Entropy | $-\sum_{x\in s} P(x) \log_2 P(x)$ |
| Peak-to-peak distance | $|\max(\boldsymbol{s}) - \min(\boldsymbol{s})|$ | Area under curve | $\sum_{i=0}^{T-1} (t_{i+1} - t_i) \cdot \frac{s_{i+1} + s_i}{2}$ |
| Absolute energy | $\sum_{i=0}^{T} s_i^2$ | Histogram | $n = \sum_{i=1}^{k} m_i$ |
| Inter-quartile range | $Q_3(\boldsymbol{s}) - Q_1(\boldsymbol{s})$ | Mean absolute deviation | $\frac{1}{T}\sum_{i=1}^{T} |s_i^2 - \text{mean}(\boldsymbol{s})|$ |
| Median absolute deviation | $\text{median}_i(|s_i - \text{median}(\boldsymbol{s})|)$ | Root mean square | $\sqrt{\frac{1}{T}\sum_{i=1}^{T} s_i^2}$ |
| Standard deviation (STD) | $\sqrt{\frac{1}{T}\sum_{i=1}^{T} (s_i - \text{mean}(\boldsymbol{s}))^2}$ | Variance (VAR) | $\frac{1}{T}\sum_{i=1}^{T} (s_i - \text{mean}(\boldsymbol{s}))^2$ |
| Wavelet absolute mean | $|\text{mean}(\text{wavelet}(\boldsymbol{s}))|$ | Wavelet standard deviation | $|\text{std}(\text{wavelet}(\boldsymbol{s}))|$ |
| Wavelet variance | $|\text{var}(\text{wavelet}(\boldsymbol{s}))|$ | Skewness | $\frac{1}{T(\text{STD})^3}\sum_{i=0}^{T} (s_i - \text{mean}(\boldsymbol{s}))^3$ |

Table 13: Feature comparison with other LTSM open source packages

| | **LTSM-Bundle** | OpenLTM | Time-LLM | LLM-Time |
|---|---|---|---|---|
| Support for multiple model architectures and prompting strategies | **Yes** | Yes | No | No |
| Integration with database | **Yes** | No | No | No |
| Data preprocessing and pipeline integration | **Yes** | No | No | No |
| Zero-shot | **Yes** | Yes | Yes | Yes |
| Visualization | **Yes** | No | No | Yes |

# E Comparison with Other Packages

In this section, we highlight the difference and advantages of `LTSM-Bundle` comparing to other existing open source LTSM packages, including OpenLTM[11], Time-LLM [12], and LLM-Time[13]. Our package involve more industrial-oriented and user-friendly features, such as database integration and report visualization.

# F Additional Experimental Results on `LTSM-Bundle`

In this section, we show additional results regarding comparing `LTSM-Bundle` with other baselines in Tables 16 and 17, results of zero-shot transfer learning in Table 14, results of different training paradigms in Table 18, results of different backbones in Table 19, results of different downsampling ratios in Table 20.

---

[11] https://github.com/thuml/OpenLTM
[12] https://github.com/KimMeen/Time-LLM
[13] https://github.com/ngruver/llmtime

### F.1   Performance Comparison with Additional Baselines

Extending the analysis presented in Section 5.2, we introduce full performance comparison with new baselines. We evaluate the proposed `LTSM-Bundle` in zero-shot and few-shot settings to highlight its efficacy and robustness in Table 16 and 17.

### F.2   Zero-shot Transfer Learning Comparisons

In addition to the results in Section 5.2, this section introduces the full zero-shot transfer learning comparisons. We evaluate the proposed `LTSM-Bundle` in the zero-shot transfer scenarios, detailed in shown in Table 14.

### F.3   Training Paradigm Comparisons

Expanding upon the results in Section 4.3, this section presents the full experimental results for the training paradigms analysis, including different backbones and prompting strategies. The analytic results are detailed in Table 18.

### F.4   Backbone Architecture Comparisons

We provide all the numbers of analytics on different backbone architectures, continuing from Section 4.4. Results are in different language model backbones, including GPT-2-Small, GPT-2-Medium, GPT-2-Large, and Phi-2, shown in Table 19.

### F.5   Down-sampling Ratio Comparisons

We here present the full version of our experimental results on the different down-sampling ratios in Section 4.6. We test `LTSM-Bundle` with GPT-Medium as backbones with the proposed TS prompt under a fully tuning paradigm. The results in the ratio of $\{40, 20, 10\}$ (i.e., downsample rate in $\{2.5\%, 5\%, 10\%\}$) are all demonstrated in Table 20.

### F.6   Different Numbers of Layer Adaptation Comparisons

We compared the average performance among all datasets of a 3-layer model and a full 24-layer model using GPT-medium as the backbone. Our results (in Table 15) show that the 24-layer model performs worse when trained with the same number of iterations. We believe this suggests that the 3-layer configuration is a reasonable strategy for benchmarking at this stage.

Table 14: Results of zero-shot transfer learning. A time-series model is trained on a source dataset and transferred to the target dataset without adaptation.

| Methods | | LTSM-Bundle | | TIME-LLM | | LLMTime | | GPT4TS | | DLinear | | PatchTST | | TimesNet | | Autoformer | |
|---|---|---|---|---|---|---|---|---|---|---|---|---|---|---|---|---|
| Metric | | MSE | MAE | MSE | MAE | MSE | MAE | MSE | MAE | MSE | MAE | MSE | MAE | MSE | MAE | MSE | MAE |
| | 96 | 0.229 | 0.326 | 0.279 | 0.337 | 0.510 | 0.576 | 0.335 | 0.374 | 0.347 | 0.400 | 0.304 | 0.350 | 0.358 | 0.387 | 0.469 | 0.486 |
| | 192 | 0.310 | 0.395 | 0.351 | 0.374 | 0.523 | 0.586 | 0.412 | 0.417 | 0.417 | 0.460 | 0.386 | 0.400 | 0.427 | 0.429 | 0.634 | 0.567 |
| ETTh1 → ETTh2 | 336 | 0.336 | 0.414 | 0.388 | 0.415 | 0.640 | 0.637 | 0.441 | 0.444 | 0.515 | 0.505 | 0.414 | 0.428 | 0.449 | 0.451 | 0.655 | 0.588 |
| | 720 | 0.401 | 0.474 | 0.391 | 0.420 | 2.296 | 1.034 | 0.438 | 0.452 | 0.665 | 0.589 | 0.419 | 0.443 | 0.448 | 0.458 | 0.570 | 0.549 |
| | Avg | 0.319 | 0.402 | 0.353 | 0.387 | 0.992 | 0.708 | 0.406 | 0.422 | 0.493 | 0.488 | 0.380 | 0.405 | 0.421 | 0.431 | 0.582 | 0.548 |
| | 96 | 0.197 | 0.318 | 0.189 | 0.293 | 0.646 | 0.563 | 0.236 | 0.315 | 0.255 | 0.357 | 0.215 | 0.304 | 0.239 | 0.313 | 0.352 | 0.432 |
| | 192 | 0.314 | 0.420 | 0.237 | 0.312 | 0.934 | 0.654 | 0.287 | 0.342 | 0.338 | 0.413 | 0.275 | 0.339 | 0.291 | 0.342 | 0.413 | 0.460 |
| ETTh1 → ETTm2 | 336 | 0.313 | 0.405 | 0.291 | 0.365 | 1.157 | 0.728 | 0.341 | 0.374 | 0.425 | 0.465 | 0.334 | 0.373 | 0.342 | 0.371 | 0.465 | 0.489 |
| | 720 | 0.425 | 0.483 | 0.372 | 0.390 | 4.730 | 1.531 | 0.435 | 0.422 | 0.640 | 0.573 | 0.431 | 0.424 | 0.434 | 0.419 | 0.599 | 0.551 |
| | Avg | 0.312 | 0.406 | 0.273 | 0.340 | 1.867 | 0.869 | 0.325 | 0.363 | 0.415 | 0.452 | 0.314 | 0.360 | 0.327 | 0.361 | 0.457 | 0.483 |
| | 96 | 0.390 | 0.439 | 0.450 | 0.452 | 1.130 | 0.777 | 0.732 | 0.577 | 0.689 | 0.555 | 0.485 | 0.465 | 0.848 | 0.601 | 0.693 | 0.569 |
| | 192 | 0.417 | 0.460 | 0.465 | 0.461 | 1.242 | 0.820 | 0.758 | 0.559 | 0.707 | 0.568 | 0.565 | 0.509 | 0.860 | 0.610 | 0.760 | 0.601 |
| ETTh2 → ETTh1 | 336 | 0.462 | 0.501 | 0.501 | 0.482 | 1.242 | 0.864 | 0.759 | 0.578 | 0.710 | 0.577 | 0.581 | 0.515 | 0.867 | 0.626 | 0.781 | 0.619 |
| | 720 | 0.568 | 0.588 | 0.501 | 0.502 | 4.145 | 1.461 | 0.781 | 0.597 | 0.704 | 0.596 | 0.628 | 0.561 | 0.887 | 0.648 | 0.796 | 0.644 |
| | Avg | 0.459 | 0.497 | 0.479 | 0.474 | 1.961 | 0.981 | 0.757 | 0.578 | 0.703 | 0.574 | 0.565 | 0.513 | 0.865 | 0.621 | 0.757 | 0.608 |
| | 96 | 0.200 | 0.316 | 0.174 | 0.276 | 0.646 | 0.563 | 0.253 | 0.329 | 0.240 | 0.336 | 0.226 | 0.309 | 0.248 | 0.324 | 0.263 | 0.352 |
| | 192 | 0.250 | 0.359 | 0.233 | 0.315 | 0.934 | 0.654 | 0.293 | 0.346 | 0.295 | 0.369 | 0.289 | 0.345 | 0.296 | 0.352 | 0.326 | 0.389 |
| ETTh2 → ETTm2 | 336 | 0.327 | 0.416 | 0.291 | 0.337 | 1.157 | 0.728 | 0.347 | 0.376 | 0.345 | 0.397 | 0.348 | 0.379 | 0.353 | 0.383 | 0.387 | 0.426 |
| | 720 | 0.573 | 0.563 | 0.392 | 0.417 | 4.730 | 1.531 | 0.446 | 0.429 | 0.432 | 0.442 | 0.439 | 0.427 | 0.471 | 0.446 | 0.487 | 0.478 |
| | Avg | 0.337 | 0.413 | 0.272 | 0.341 | 1.867 | 0.869 | 0.335 | 0.370 | 0.328 | 0.386 | 0.325 | 0.365 | 0.342 | 0.376 | 0.366 | 0.411 |
| | 96 | 0.246 | 0.342 | 0.321 | 0.369 | 0.510 | 0.576 | 0.353 | 0.392 | 0.365 | 0.415 | 0.354 | 0.385 | 0.377 | 0.407 | 0.435 | 0.470 |
| | 192 | 0.290 | 0.374 | 0.389 | 0.410 | 0.523 | 0.586 | 0.443 | 0.437 | 0.454 | 0.462 | 0.447 | 0.434 | 0.471 | 0.453 | 0.495 | 0.489 |
| ETTm1 → ETTh2 | 336 | 0.326 | 0.406 | 0.408 | 0.433 | 0.640 | 0.637 | 0.469 | 0.461 | 0.496 | 0.464 | 0.481 | 0.463 | 0.472 | 0.484 | 0.470 | 0.472 |
| | 720 | 0.363 | 0.440 | 0.406 | 0.436 | 2.296 | 1.034 | 0.466 | 0.468 | 0.541 | 0.529 | 0.474 | 0.471 | 0.495 | 0.482 | 0.480 | 0.485 |
| | Avg | 0.306 | 0.391 | 0.381 | 0.412 | 0.992 | 0.708 | 0.433 | 0.439 | 0.464 | 0.475 | 0.439 | 0.438 | 0.457 | 0.454 | 0.470 | 0.479 |
| | 96 | 0.144 | 0.257 | 0.169 | 0.257 | 0.646 | 0.563 | 0.217 | 0.294 | 0.221 | 0.314 | 0.195 | 0.271 | 0.222 | 0.295 | 0.385 | 0.457 |
| | 192 | 0.193 | 0.302 | 0.227 | 0.318 | 0.934 | 0.654 | 0.277 | 0.327 | 0.286 | 0.359 | 0.258 | 0.311 | 0.288 | 0.337 | 0.433 | 0.469 |
| ETTm1 → ETTm2 | 336 | 0.240 | 0.342 | 0.290 | 0.338 | 1.157 | 0.728 | 0.331 | 0.360 | 0.357 | 0.406 | 0.317 | 0.348 | 0.341 | 0.367 | 0.476 | 0.477 |
| | 720 | 0.292 | 0.379 | 0.375 | 0.367 | 4.730 | 1.531 | 0.429 | 0.413 | 0.476 | 0.476 | 0.416 | 0.404 | 0.436 | 0.418 | 0.582 | 0.535 |
| | Avg | 0.217 | 0.320 | 0.268 | 0.320 | 1.867 | 0.869 | 0.313 | 0.348 | 0.335 | 0.389 | 0.296 | 0.334 | 0.322 | 0.354 | 0.469 | 0.484 |
| | 96 | 0.257 | 0.346 | 0.298 | 0.356 | 0.510 | 0.576 | 0.360 | 0.401 | 0.333 | 0.391 | 0.327 | 0.367 | 0.360 | 0.401 | 0.353 | 0.393 |
| | 192 | 0.309 | 0.382 | 0.359 | 0.397 | 0.523 | 0.586 | 0.434 | 0.437 | 0.441 | 0.456 | 0.411 | 0.418 | 0.434 | 0.437 | 0.432 | 0.437 |
| ETTm2 → ETTh2 | 336 | 0.341 | 0.413 | 0.367 | 0.412 | 0.640 | 0.637 | 0.460 | 0.459 | 0.505 | 0.503 | 0.439 | 0.447 | 0.460 | 0.459 | 0.452 | 0.459 |
| | 720 | 0.350 | 0.432 | 0.393 | 0.434 | 2.296 | 1.034 | 0.485 | 0.477 | 0.543 | 0.534 | 0.459 | 0.470 | 0.485 | 0.477 | 0.453 | 0.467 |
| | Avg | 0.314 | 0.393 | 0.354 | 0.400 | 0.992 | 0.708 | 0.435 | 0.443 | 0.455 | 0.471 | 0.409 | 0.425 | 0.435 | 0.443 | 0.423 | 0.439 |
| | 96 | 0.364 | 0.410 | 0.359 | 0.397 | 1.179 | 0.781 | 0.747 | 0.558 | 0.570 | 0.490 | 0.491 | 0.437 | 0.747 | 0.558 | 0.735 | 0.576 |
| | 192 | 0.405 | 0.432 | 0.390 | 0.420 | 1.327 | 0.846 | 0.781 | 0.560 | 0.590 | 0.506 | 0.530 | 0.470 | 0.781 | 0.560 | 0.753 | 0.586 |
| ETTm2 → ETTm1 | 336 | 0.413 | 0.433 | 0.421 | 0.445 | 1.478 | 0.902 | 0.778 | 0.578 | 0.706 | 0.567 | 0.565 | 0.497 | 0.778 | 0.578 | 0.750 | 0.593 |
| | 720 | 0.432 | 0.446 | 0.487 | 0.488 | 3.749 | 1.408 | 0.769 | 0.573 | 0.731 | 0.584 | 0.686 | 0.565 | 0.769 | 0.573 | 0.782 | 0.609 |
| | Avg | 0.403 | 0.430 | 0.414 | 0.438 | 1.933 | 0.984 | 0.769 | 0.567 | 0.649 | 0.537 | 0.568 | 0.492 | 0.769 | 0.667 | 0.755 | 0.591 |

Table 15: Performance comparison between 3-layer and 24-layer of LTSM-Bundle.

| | 3-layer | 24-layer |
|---|---|---|
| MAE | 0.2003 | 0.2439 |
| MSE | 0.2770 | 0.3162 |

Table 16: Performance comparison with additional baselines (Full data)

| Methods | | LTSM-Bundle | | TIME-LLM | | TEST | | LLM4TS | | GPT4TS | | DLinear | | PatchTST | | TimesNet | | FEDformer | | Autoformer | | Non-Stationary | | ETSformer | | LightTS | | Informer | | Reformer | |
|---|---|---|---|---|---|---|---|---|---|---|---|---|---|---|---|---|---|---|---|---|---|---|---|---|---|---|---|---|---|---|---|---|
| Metric | | MSE | MAE | MSE | MAE | MSE | MAE | MSE | MAE | MSE | MAE | MSE | MAE | MSE | MAE | MSE | MAE | MSE | MAE | MSE | MAE | MSE | MAE | MSE | MAE | MSE | MAE | MSE | MAE | MSE | MAE |
| ETTh1 | 96 | 0.307 | 0.377 | 0.362 | 0.392 | 0.372 | 0.400 | 0.371 | 0.394 | 0.376 | 0.397 | 0.375 | 0.399 | 0.370 | 0.399 | 0.384 | 0.402 | 0.376 | 0.419 | 0.449 | 0.459 | 0.513 | 0.491 | 0.494 | 0.479 | 0.424 | 0.432 | 0.865 | 0.713 | 0.837 | 0.728 |
| | 192 | 0.329 | 0.391 | 0.398 | 0.418 | 0.414 | 0.422 | 0.403 | 0.412 | 0.416 | 0.418 | 0.405 | 0.416 | 0.413 | 0.421 | 0.436 | 0.429 | 0.420 | 0.448 | 0.500 | 0.482 | 0.534 | 0.504 | 0.538 | 0.504 | 0.475 | 0.462 | 1.008 | 0.792 | 0.923 | 0.766 |
| | 336 | 0.346 | 0.405 | 0.430 | 0.427 | 0.422 | 0.437 | 0.420 | 0.422 | 0.442 | 0.433 | 0.439 | 0.443 | 0.422 | 0.436 | 0.491 | 0.469 | 0.459 | 0.465 | 0.521 | 0.496 | 0.588 | 0.535 | 0.574 | 0.521 | 0.518 | 0.488 | 1.107 | 0.809 | 1.097 | 0.835 |
| | 720 | 0.370 | 0.441 | 0.442 | 0.457 | 0.447 | 0.467 | 0.422 | 0.444 | 0.477 | 0.456 | 0.472 | 0.490 | 0.447 | 0.466 | 0.521 | 0.500 | 0.506 | 0.507 | 0.514 | 0.512 | 0.643 | 0.616 | 0.562 | 0.535 | 0.547 | 0.533 | 1.181 | 0.865 | 1.257 | 0.889 |
| | Avg | 0.338 | 0.403 | 0.408 | 0.423 | 0.414 | 0.431 | 0.404 | 0.418 | 0.465 | 0.455 | 0.422 | 0.437 | 0.413 | 0.430 | 0.458 | 0.450 | 0.440 | 0.460 | 0.496 | 0.487 | 0.570 | 0.537 | 0.542 | 0.510 | 0.491 | 0.479 | 1.040 | 0.795 | 1.029 | 0.805 |
| ETTh2 | 96 | 0.235 | 0.326 | 0.268 | 0.328 | 0.275 | 0.338 | 0.269 | 0.332 | 0.285 | 0.342 | 0.289 | 0.353 | 0.274 | 0.336 | 0.340 | 0.374 | 0.358 | 0.397 | 0.346 | 0.388 | 0.476 | 0.458 | 0.340 | 0.391 | 0.397 | 0.437 | 3.755 | 1.525 | 2.626 | 1.317 |
| | 192 | 0.283 | 0.365 | 0.329 | 0.375 | 0.340 | 0.379 | 0.328 | 0.377 | 0.354 | 0.389 | 0.383 | 0.418 | 0.339 | 0.379 | 0.402 | 0.414 | 0.429 | 0.439 | 0.456 | 0.452 | 0.512 | 0.493 | 0.430 | 0.439 | 0.520 | 0.504 | 5.602 | 1.931 | 11.12 | 2.979 |
| | 336 | 0.320 | 0.401 | 0.368 | 0.409 | 0.329 | 0.381 | 0.353 | 0.369 | 0.373 | 0.407 | 0.448 | 0.465 | 0.329 | 0.380 | 0.452 | 0.452 | 0.496 | 0.487 | 0.482 | 0.486 | 0.552 | 0.551 | 0.485 | 0.479 | 0.626 | 0.559 | 4.721 | 1.835 | 9.323 | 2.769 |
| | 720 | 0.378 | 0.456 | 0.372 | 0.420 | 0.381 | 0.423 | 0.383 | 0.425 | 0.406 | 0.441 | 0.605 | 0.551 | 0.379 | 0.422 | 0.462 | 0.468 | 0.463 | 0.474 | 0.515 | 0.511 | 0.562 | 0.560 | 0.500 | 0.497 | 0.863 | 0.672 | 3.647 | 1.625 | 3.874 | 1.697 |
| | Avg | 0.304 | 0.387 | 0.334 | 0.383 | 0.331 | 0.380 | 0.333 | 0.376 | 0.381 | 0.412 | 0.431 | 0.446 | 0.330 | 0.379 | 0.414 | 0.427 | 0.437 | 0.449 | 0.450 | 0.459 | 0.526 | 0.516 | 0.439 | 0.452 | 0.602 | 0.543 | 4.431 | 1.729 | 6.736 | 2.191 |
| ETTm1 | 96 | 0.285 | 0.369 | 0.272 | 0.334 | 0.293 | 0.346 | 0.285 | 0.343 | 0.292 | 0.346 | 0.299 | 0.343 | 0.290 | 0.342 | 0.338 | 0.375 | 0.379 | 0.419 | 0.505 | 0.475 | 0.386 | 0.398 | 0.375 | 0.398 | 0.374 | 0.400 | 0.672 | 0.571 | 0.538 | 0.528 |
| | 192 | 0.319 | 0.393 | 0.310 | 0.358 | 0.332 | 0.369 | 0.324 | 0.366 | 0.332 | 0.372 | 0.335 | 0.365 | 0.332 | 0.369 | 0.374 | 0.387 | 0.426 | 0.441 | 0.553 | 0.496 | 0.459 | 0.444 | 0.408 | 0.410 | 0.400 | 0.407 | 0.795 | 0.669 | 0.658 | 0.592 |
| | 336 | 0.378 | 0.425 | 0.352 | 0.384 | 0.368 | 0.392 | 0.353 | 0.385 | 0.366 | 0.394 | 0.369 | 0.386 | 0.366 | 0.392 | 0.410 | 0.411 | 0.445 | 0.459 | 0.621 | 0.537 | 0.495 | 0.464 | 0.435 | 0.428 | 0.438 | 0.438 | 1.212 | 0.871 | 0.898 | 0.721 |
| | 720 | 0.464 | 0.477 | 0.383 | 0.411 | 0.418 | 0.420 | 0.408 | 0.419 | 0.417 | 0.421 | 0.425 | 0.421 | 0.416 | 0.420 | 0.478 | 0.450 | 0.543 | 0.490 | 0.671 | 0.561 | 0.585 | 0.516 | 0.499 | 0.462 | 0.527 | 0.502 | 1.166 | 0.823 | 1.102 | 0.841 |
| | Avg | 0.362 | 0.416 | 0.329 | 0.372 | 0.353 | 0.382 | 0.343 | 0.378 | 0.388 | 0.403 | 0.357 | 0.378 | 0.351 | 0.380 | 0.400 | 0.406 | 0.448 | 0.452 | 0.588 | 0.517 | 0.481 | 0.456 | 0.429 | 0.425 | 0.435 | 0.437 | 0.961 | 0.734 | 0.799 | 0.671 |
| ETTm2 | 96 | 0.156 | 0.266 | 0.161 | 0.253 | - | - | 0.165 | 0.254 | 0.173 | 0.262 | 0.167 | 0.269 | 0.165 | 0.255 | 0.187 | 0.267 | 0.203 | 0.287 | 0.255 | 0.339 | 0.192 | 0.274 | 0.189 | 0.280 | 0.209 | 0.308 | 0.365 | 0.453 | 0.658 | 0.619 |
| | 192 | 0.203 | 0.307 | 0.219 | 0.293 | - | - | 0.220 | 0.292 | 0.229 | 0.301 | 0.224 | 0.303 | 0.220 | 0.292 | 0.249 | 0.309 | 0.269 | 0.328 | 0.281 | 0.340 | 0.280 | 0.339 | 0.253 | 0.319 | 0.311 | 0.382 | 0.533 | 0.563 | 1.078 | 0.827 |
| | 336 | 0.255 | 0.349 | 0.271 | 0.329 | - | - | 0.268 | 0.326 | 0.286 | 0.341 | 0.281 | 0.342 | 0.274 | 0.329 | 0.321 | 0.351 | 0.325 | 0.366 | 0.339 | 0.372 | 0.334 | 0.361 | 0.314 | 0.357 | 0.442 | 0.466 | 1.363 | 0.887 | 1.549 | 0.972 |
| | 720 | 0.342 | 0.417 | 0.352 | 0.379 | - | - | 0.350 | 0.380 | 0.378 | 0.401 | 0.397 | 0.421 | 0.362 | 0.385 | 0.408 | 0.403 | 0.421 | 0.415 | 0.433 | 0.432 | 0.417 | 0.413 | 0.414 | 0.413 | 0.675 | 0.587 | 3.379 | 1.338 | 2.631 | 1.242 |
| | Avg | 0.239 | 0.335 | 0.251 | 0.313 | - | - | 0.251 | 0.313 | 0.284 | 0.339 | 0.267 | 0.333 | 0.255 | 0.315 | 0.291 | 0.333 | 0.305 | 0.349 | 0.327 | 0.371 | 0.306 | 0.347 | 0.293 | 0.342 | 0.409 | 0.436 | 1.410 | 0.810 | 1.479 | 0.915 |
| Weather | 96 | 0.172 | 0.242 | 0.147 | 0.201 | 0.150 | 0.202 | 0.147 | 0.196 | 0.162 | 0.212 | 0.176 | 0.237 | 0.149 | 0.198 | 0.172 | 0.220 | 0.217 | 0.296 | 0.266 | 0.336 | 0.173 | 0.223 | 0.197 | 0.281 | 0.182 | 0.242 | 0.300 | 0.384 | 0.689 | 0.596 |
| | 192 | 0.218 | 0.278 | 0.189 | 0.234 | 0.198 | 0.246 | 0.191 | 0.238 | 0.204 | 0.248 | 0.220 | 0.282 | 0.194 | 0.241 | 0.219 | 0.261 | 0.276 | 0.336 | 0.307 | 0.367 | 0.245 | 0.285 | 0.237 | 0.312 | 0.227 | 0.287 | 0.598 | 0.544 | 0.752 | 0.638 |
| | 336 | 0.276 | 0.329 | 0.262 | 0.279 | 0.245 | 0.286 | 0.241 | 0.277 | 0.254 | 0.286 | 0.265 | 0.319 | 0.245 | 0.282 | 0.280 | 0.306 | 0.339 | 0.380 | 0.359 | 0.395 | 0.321 | 0.338 | 0.298 | 0.353 | 0.282 | 0.334 | 0.578 | 0.523 | 0.639 | 0.596 |
| | 720 | 0.339 | 0.373 | 0.304 | 0.316 | 0.324 | 0.342 | 0.313 | 0.329 | 0.326 | 0.337 | 0.333 | 0.362 | 0.314 | 0.334 | 0.365 | 0.359 | 0.403 | 0.428 | 0.419 | 0.428 | 0.414 | 0.410 | 0.352 | 0.288 | 0.352 | 0.386 | 1.059 | 0.741 | 1.130 | 0.792 |
| | Avg | 0.251 | 0.305 | 0.225 | 0.257 | 0.229 | 0.271 | 0.223 | 0.260 | 0.237 | 0.270 | 0.248 | 0.300 | 0.225 | 0.264 | 0.259 | 0.287 | 0.309 | 0.360 | 0.338 | 0.382 | 0.288 | 0.314 | 0.271 | 0.334 | 0.261 | 0.312 | 0.634 | 0.548 | 0.803 | 0.656 |
| Electricity | 96 | 0.145 | 0.247 | 0.131 | 0.224 | 0.132 | 0.223 | 0.128 | 0.223 | 0.139 | 0.238 | 0.140 | 0.237 | 0.129 | 0.222 | 0.168 | 0.272 | 0.193 | 0.308 | 0.201 | 0.317 | 0.169 | 0.273 | 0.187 | 0.304 | 0.207 | 0.307 | 0.274 | 0.368 | 0.312 | 0.402 |
| | 192 | 0.159 | 0.259 | 0.152 | 0.241 | 0.158 | 0.241 | 0.146 | 0.240 | 0.153 | 0.251 | 0.153 | 0.249 | 0.157 | 0.240 | 0.184 | 0.289 | 0.201 | 0.315 | 0.222 | 0.334 | 0.182 | 0.286 | 0.199 | 0.315 | 0.213 | 0.316 | 0.296 | 0.386 | 0.348 | 0.433 |
| | 336 | 0.180 | 0.284 | 0.160 | 0.248 | 0.163 | 0.260 | 0.163 | 0.258 | 0.169 | 0.266 | 0.169 | 0.267 | 0.163 | 0.259 | 0.198 | 0.300 | 0.214 | 0.329 | 0.231 | 0.338 | 0.200 | 0.304 | 0.212 | 0.329 | 0.230 | 0.333 | 0.300 | 0.394 | 0.350 | 0.433 |
| | 720 | 0.215 | 0.317 | 0.192 | 0.298 | 0.199 | 0.291 | 0.200 | 0.292 | 0.206 | 0.297 | 0.203 | 0.301 | 0.197 | 0.290 | 0.220 | 0.320 | 0.246 | 0.355 | 0.254 | 0.361 | 0.222 | 0.321 | 0.233 | 0.345 | 0.265 | 0.360 | 0.373 | 0.439 | 0.340 | 0.420 |
| | Avg | 0.175 | 0.276 | 0.158 | 0.252 | 0.162 | 0.253 | 0.159 | 0.253 | 0.167 | 0.263 | 0.166 | 0.263 | 0.161 | 0.252 | 0.192 | 0.295 | 0.214 | 0.327 | 0.227 | 0.338 | 0.193 | 0.296 | 0.208 | 0.323 | 0.229 | 0.329 | 0.311 | 0.397 | 0.338 | 0.422 |
| Traffic | 96 | 0.305 | 0.279 | 0.362 | 0.248 | 0.407 | 0.282 | 0.372 | 0.259 | 0.388 | 0.282 | 0.410 | 0.282 | 0.360 | 0.249 | 0.593 | 0.321 | 0.587 | 0.366 | 0.613 | 0.388 | 0.612 | 0.338 | 0.607 | 0.392 | 0.615 | 0.391 | 0.719 | 0.391 | 0.732 | 0.423 |
| | 192 | 0.313 | 0.274 | 0.374 | 0.247 | 0.423 | 0.287 | 0.391 | 0.265 | 0.407 | 0.290 | 0.423 | 0.287 | 0.379 | 0.256 | 0.617 | 0.336 | 0.604 | 0.373 | 0.616 | 0.382 | 0.613 | 0.340 | 0.621 | 0.399 | 0.601 | 0.382 | 0.696 | 0.379 | 0.733 | 0.420 |
| | 336 | 0.326 | 0.287 | 0.385 | 0.271 | 0.430 | 0.296 | 0.405 | 0.275 | 0.412 | 0.294 | 0.436 | 0.296 | 0.392 | 0.264 | 0.629 | 0.336 | 0.621 | 0.383 | 0.622 | 0.337 | 0.618 | 0.328 | 0.622 | 0.396 | 0.613 | 0.386 | 0.777 | 0.420 | 0.742 | 0.420 |
| | 720 | 0.346 | 0.301 | 0.430 | 0.288 | 0.463 | 0.315 | 0.437 | 0.292 | 0.450 | 0.312 | 0.466 | 0.315 | 0.432 | 0.286 | 0.640 | 0.350 | 0.626 | 0.382 | 0.660 | 0.408 | 0.653 | 0.355 | 0.632 | 0.396 | 0.658 | 0.407 | 0.864 | 0.472 | 0.755 | 0.423 |
| | Avg | 0.323 | 0.285 | 0.388 | 0.264 | 0.430 | 0.295 | 0.401 | 0.273 | 0.414 | 0.294 | 0.433 | 0.295 | 0.390 | 0.263 | 0.620 | 0.336 | 0.610 | 0.376 | 0.628 | 0.379 | 0.624 | 0.340 | 0.621 | 0.396 | 0.622 | 0.392 | 0.764 | 0.416 | 0.741 | 0.422 |

Table 17: Performance comparison with additional baselines (5% Few Shot data)

| Methods | | LTSM-Bundle | | TIME-LLM | | LLM4TS | | GPT4TS | | DLinear | | PatchTST | | TimesNet | | FEDformer | | Autoformer | | Non-Stationary | | ETSformer | | LightTS | | Informer | | Reformer | |
|---|---|---|---|---|---|---|---|---|---|---|---|---|---|---|---|---|---|---|---|---|---|---|---|---|---|---|---|---|---|
| Metric | | MSE | MAE | MSE | MAE | MSE | MAE | MSE | MAE | MSE | MAE | MSE | MAE | MSE | MAE | MSE | MAE | MSE | MAE | MSE | MAE | MSE | MAE | MSE | MAE | MSE | MAE | MSE | MAE |
| ETTh1 | 96 | 0.307 | 0.377 | 0.483 | 0.464 | 0.509 | 0.484 | 0.543 | 0.506 | 0.547 | 0.503 | 0.557 | 0.519 | 0.892 | 0.62 | 0.593 | 0.529 | 0.681 | 0.570 | 0.952 | 0.650 | 1.169 | 0.832 | 1.483 | 0.91 | 1.225 | 0.812 | 1.198 | 0.795 |
| | 192 | 0.329 | 0.391 | 0.629 | 0.540 | 0.717 | 0.581 | 0.748 | 0.580 | 0.720 | 0.604 | 0.711 | 0.570 | 0.940 | 0.665 | 0.652 | 0.563 | 0.725 | 0.602 | 0.943 | 0.645 | 1.221 | 0.853 | 1.525 | 0.93 | 1.249 | 0.828 | 1.273 | 0.853 |
| | 336 | 0.346 | 0.405 | 0.768 | 0.626 | 0.728 | 0.589 | 0.754 | 0.595 | 0.984 | 0.727 | 0.816 | 0.619 | 0.945 | 0.653 | 0.731 | 0.594 | 0.761 | 0.624 | 0.935 | 0.644 | 1.179 | 0.832 | 1.347 | 0.87 | 1.202 | 0.811 | 1.254 | 0.857 |
| | 720 | 0.370 | 0.441 | - | - | - | - | - | - | - | - | - | - | - | - | - | - | - | - | - | - | - | - | - | - | - | - | - | - |
| | Avg | 0.338 | 0.403 | 0.627 | 0.543 | 0.651 | 0.551 | 0.681 | 0.560 | 0.750 | 0.611 | 0.694 | 0.569 | 0.925 | 0.647 | 0.658 | 0.562 | 0.722 | 0.598 | 0.943 | 0.646 | 1.189 | 0.839 | 1.451 | 0.903 | 1.225 | 0.817 | 1.241 | 0.835 |
| ETTh2 | 96 | 0.235 | 0.326 | 0.336 | 0.397 | 0.314 | 0.375 | 0.376 | 0.421 | 0.442 | 0.456 | 0.401 | 0.421 | 0.409 | 0.420 | 0.390 | 0.424 | 0.428 | 0.468 | 0.408 | 0.423 | 0.678 | 0.619 | 2.022 | 1.006 | 3.837 | 1.508 | 3.753 | 1.518 |
| | 192 | 0.283 | 0.365 | 0.406 | 0.425 | 0.365 | 0.408 | 0.418 | 0.441 | 0.617 | 0.542 | 0.452 | 0.455 | 0.483 | 0.464 | 0.457 | 0.465 | 0.496 | 0.504 | 0.497 | 0.468 | 0.845 | 0.697 | 3.534 | 1.348 | 3.975 | 1.933 | 3.516 | 1.473 |
| | 336 | 0.320 | 0.401 | 0.405 | 0.432 | 0.398 | 0.432 | 0.408 | 0.439 | 1.424 | 0.849 | 0.464 | 0.469 | 0.499 | 0.479 | 0.477 | 0.483 | 0.486 | 0.496 | 0.507 | 0.481 | 0.905 | 0.727 | 4.063 | 1.451 | 3.956 | 1.520 | 3.312 | 1.427 |
| | 720 | 0.378 | 0.456 | - | - | - | - | - | - | - | - | - | - | - | - | - | - | - | - | - | - | - | - | - | - | - | - | - | - |
| | Avg | 0.304 | 0.387 | 0.382 | 0.418 | 0.359 | 0.405 | 0.400 | 0.433 | 0.694 | 0.577 | 0.439 | 0.448 | 0.439 | 0.448 | 0.463 | 0.454 | 0.441 | 0.457 | 0.470 | 0.489 | 0.809 | 0.681 | 3.206 | 1.268 | 3.922 | 1.653 | 3.527 | 1.472 |
| ETTm1 | 96 | 0.285 | 0.369 | 0.316 | 0.377 | 0.349 | 0.379 | 0.386 | 0.405 | 0.332 | 0.374 | 0.399 | 0.414 | 0.606 | 0.518 | 0.628 | 0.544 | 0.726 | 0.578 | 0.823 | 0.587 | 1.031 | 0.747 | 1.048 | 0.733 | 1.130 | 0.775 | 1.234 | 0.798 |
| | 192 | 0.319 | 0.393 | 0.450 | 0.464 | 0.374 | 0.394 | 0.440 | 0.438 | 0.358 | 0.390 | 0.441 | 0.436 | 0.681 | 0.539 | 0.666 | 0.566 | 0.750 | 0.591 | 0.844 | 0.591 | 1.087 | 0.766 | 1.097 | 0.756 | 1.150 | 0.788 | 1.287 | 0.839 |
| | 336 | 0.378 | 0.425 | 0.450 | 0.424 | 0.411 | 0.417 | 0.485 | 0.459 | 0.402 | 0.416 | 0.499 | 0.467 | 0.786 | 0.597 | 0.807 | 0.628 | 0.851 | 0.659 | 0.870 | 0.603 | 1.138 | 0.787 | 1.147 | 0.775 | 1.198 | 0.809 | 1.288 | 0.842 |
| | 720 | 0.464 | 0.477 | 0.483 | 0.471 | 0.516 | 0.479 | 0.577 | 0.499 | 0.511 | 0.489 | 0.767 | 0.587 | 0.796 | 0.593 | 0.822 | 0.633 | 0.857 | 0.655 | 0.893 | 0.611 | 1.245 | 0.831 | 1.200 | 0.799 | 1.175 | 0.794 | 1.247 | 0.828 |
| | Avg | 0.362 | 0.416 | 0.425 | 0.434 | 0.412 | 0.417 | 0.472 | 0.450 | 0.400 | 0.417 | 0.526 | 0.476 | 0.717 | 0.561 | 0.730 | 0.592 | 0.796 | 0.620 | 0.857 | 0.598 | 1.125 | 0.782 | 1.123 | 0.765 | 1.163 | 0.791 | 1.264 | 0.826 |
| ETTm2 | 96 | 0.156 | 0.266 | 0.174 | 0.261 | 0.192 | 0.273 | 0.199 | 0.280 | 0.236 | 0.326 | 0.206 | 0.288 | 0.220 | 0.299 | 0.229 | 0.320 | 0.232 | 0.322 | 0.238 | 0.316 | 0.404 | 0.485 | 1.108 | 0.772 | 3.599 | 1.478 | 3.883 | 1.545 |
| | 192 | 0.203 | 0.307 | 0.215 | 0.287 | 0.249 | 0.309 | 0.256 | 0.316 | 0.306 | 0.373 | 0.264 | 0.324 | 0.311 | 0.361 | 0.394 | 0.361 | 0.291 | 0.357 | 0.298 | 0.349 | 0.479 | 0.521 | 1.317 | 0.850 | 3.578 | 1.475 | 3.553 | 1.484 |
| | 336 | 0.255 | 0.349 | 0.273 | 0.330 | 0.301 | 0.342 | 0.318 | 0.353 | 0.380 | 0.423 | 0.334 | 0.367 | 0.338 | 0.366 | 0.378 | 0.427 | 0.478 | 0.517 | 0.353 | 0.380 | 0.552 | 0.555 | 1.415 | 0.879 | 3.561 | 1.473 | 3.446 | 1.460 |
| | 720 | 0.342 | 0.417 | 0.433 | 0.412 | 0.402 | 0.405 | 0.460 | 0.436 | 0.674 | 0.583 | 0.454 | 0.432 | 0.509 | 0.465 | 0.523 | 0.510 | 0.553 | 0.538 | 0.475 | 0.445 | 0.701 | 0.627 | 1.822 | 0.984 | 3.896 | 1.533 | 3.445 | 1.460 |
| | Avg | 0.239 | 0.335 | 0.274 | 0.323 | 0.286 | 0.332 | 0.308 | 0.346 | 0.399 | 0.426 | 0.314 | 0.352 | 0.344 | 0.372 | 0.381 | 0.404 | 0.388 | 0.433 | 0.341 | 0.372 | 0.534 | 0.547 | 1.415 | 0.871 | 3.658 | 1.489 | 3.581 | 1.487 |
| Weather | 96 | 0.172 | 0.242 | 0.172 | 0.263 | 0.173 | 0.227 | 0.175 | 0.230 | 0.184 | 0.242 | 0.171 | 0.224 | 0.207 | 0.253 | 0.229 | 0.309 | 0.227 | 0.299 | 0.215 | 0.252 | 0.218 | 0.295 | 0.230 | 0.285 | 0.497 | 0.497 | 0.406 | 0.435 |
| | 192 | 0.218 | 0.278 | 0.224 | 0.271 | 0.218 | 0.265 | 0.227 | 0.276 | 0.228 | 0.283 | 0.230 | 0.277 | 0.272 | 0.307 | 0.265 | 0.317 | 0.278 | 0.333 | 0.290 | 0.307 | 0.294 | 0.331 | 0.274 | 0.323 | 0.620 | 0.545 | 0.446 | 0.450 |
| | 336 | 0.276 | 0.329 | 0.282 | 0.321 | 0.276 | 0.310 | 0.286 | 0.322 | 0.279 | 0.322 | 0.294 | 0.326 | 0.313 | 0.328 | 0.353 | 0.392 | 0.351 | 0.393 | 0.353 | 0.348 | 0.359 | 0.398 | 0.318 | 0.355 | 0.649 | 0.547 | 0.465 | 0.459 |
| | 720 | 0.339 | 0.373 | 0.366 | 0.381 | 0.355 | 0.366 | 0.366 | 0.379 | 0.364 | 0.388 | 0.384 | 0.387 | 0.400 | 0.385 | 0.391 | 0.394 | 0.387 | 0.389 | 0.452 | 0.407 | 0.461 | 0.461 | 0.401 | 0.418 | 0.570 | 0.522 | 0.471 | 0.468 |
| | Avg | 0.251 | 0.305 | 0.260 | 0.309 | 0.251 | 0.292 | 0.263 | 0.301 | 0.263 | 0.308 | 0.269 | 0.303 | 0.298 | 0.318 | 0.309 | 0.353 | 0.310 | 0.353 | 0.327 | 0.328 | 0.333 | 0.371 | 0.305 | 0.345 | 0.584 | 0.527 | 0.447 | 0.453 |
| Electricity | 96 | 0.145 | 0.247 | 0.147 | 0.242 | 0.139 | 0.235 | 0.143 | 0.241 | 0.150 | 0.251 | 0.145 | 0.244 | 0.315 | 0.389 | 0.235 | 0.322 | 0.297 | 0.367 | 0.484 | 0.518 | 0.697 | 0.638 | 0.639 | 0.609 | 1.265 | 0.919 | 1.414 | 0.855 |
| | 192 | 0.159 | 0.259 | 0.158 | 0.241 | 0.155 | 0.249 | 0.159 | 0.255 | 0.163 | 0.263 | 0.163 | 0.260 | 0.318 | 0.396 | 0.247 | 0.341 | 0.308 | 0.375 | 0.501 | 0.531 | 0.718 | 0.648 | 0.772 | 0.678 | 1.298 | 0.939 | 1.240 | 0.919 |
| | 336 | 0.180 | 0.284 | 0.178 | 0.277 | 0.174 | 0.269 | 0.179 | 0.274 | 0.175 | 0.278 | 0.183 | 0.281 | 0.340 | 0.415 | 0.267 | 0.356 | 0.354 | 0.411 | 0.574 | 0.578 | 0.758 | 0.667 | 0.901 | 0.745 | 1.302 | 0.942 | 1.253 | 0.921 |
| | 720 | 0.215 | 0.317 | 0.224 | 0.312 | 0.222 | 0.310 | 0.233 | 0.323 | 0.219 | 0.311 | 0.233 | 0.323 | 0.635 | 0.613 | 0.318 | 0.394 | 0.426 | 0.466 | 0.952 | 0.786 | 1.028 | 0.788 | 1.200 | 0.871 | 1.259 | 0.919 | 1.249 | 0.921 |
| | Avg | 0.175 | 0.276 | 0.179 | 0.268 | 0.173 | 0.266 | 0.178 | 0.273 | 0.176 | 0.275 | 0.181 | 0.277 | 0.402 | 0.453 | 0.266 | 0.353 | 0.346 | 0.404 | 0.627 | 0.603 | 0.800 | 0.685 | 0.878 | 0.725 | 1.281 | 0.929 | 1.289 | 0.904 |
| Traffic | 96 | 0.305 | 0.279 | 0.414 | 0.291 | 0.401 | 0.285 | 0.419 | 0.298 | 0.427 | 0.304 | 0.404 | 0.286 | 0.854 | 0.492 | 0.670 | 0.421 | 0.795 | 0.481 | 1.468 | 0.821 | 1.643 | 0.855 | 1.157 | 0.636 | 1.557 | 0.821 | 1.586 | 0.841 |
| | 192 | 0.313 | 0.274 | 0.419 | 0.291 | 0.418 | 0.293 | 0.434 | 0.305 | 0.447 | 0.315 | 0.412 | 0.294 | 0.894 | 0.517 | 0.653 | 0.405 | 0.837 | 0.503 | 1.509 | 0.838 | 1.856 | 0.928 | 1.688 | 0.848 | 1.596 | 0.834 | 1.602 | 0.844 |
| | 336 | 0.326 | 0.287 | 0.437 | 0.314 | 0.436 | 0.308 | 0.449 | 0.313 | 0.478 | 0.333 | 0.439 | 0.310 | 0.853 | 0.471 | 0.707 | 0.445 | 0.867 | 0.523 | 1.602 | 0.860 | 2.080 | 0.999 | 1.826 | 0.903 | 1.621 | 0.841 | 1.668 | 0.868 |
| | 720 | 0.346 | 0.301 | - | - | - | - | - | - | - | - | - | - | - | - | - | - | - | - | - | - | - | - | - | - | - | - | - | - |
| | Avg | 0.323 | 0.285 | 0.423 | 0.298 | 0.418 | 0.295 | 0.434 | 0.305 | 0.450 | 0.317 | 0.418 | 0.296 | 0.867 | 0.493 | 0.676 | 0.423 | 0.833 | 0.502 | 1.526 | 0.839 | 1.859 | 0.927 | 1.557 | 0.795 | 1.591 | 0.832 | 1.618 | 0.851 |

Table 18: Results of different backbones, training paradigms, and prompting strategies.

| Datasets | | ETTh1 | | ETTh2 | | ETTm1 | | ETTm2 | | Traffic | | Weather | | Exchange | | ECL | |
| --- | --- | --- | --- | --- | --- | --- | --- | --- | --- | --- | --- | --- | --- | --- | --- | --- | --- |
| Metric | | MSE | MAE | MSE | MAE | MSE | MAE | MSE | MAE | MSE | MAE | MSE | MAE | MSE | MAE | MSE | MAE |
| From scratch + GPT-Medium + TS prompt | 96 | 0.354 | 0.415 | 0.259 | 0.350 | 0.551 | 0.507 | 0.215 | 0.319 | 0.393 | 0.377 | 0.222 | 0.292 | 0.108 | 0.246 | 0.250 | 0.342 |
| | 192 | 0.364 | 0.421 | 0.537 | 0.505 | 0.231 | 0.331 | 0.235 | 0.335 | 0.373 | 0.356 | 0.246 | 0.309 | 0.143 | 0.288 | 0.231 | 0.331 |
| | 336 | 0.359 | 0.420 | 0.321 | 0.402 | 0.423 | 0.454 | 0.267 | 0.360 | 0.357 | 0.329 | 0.283 | 0.335 | 0.207 | 0.344 | 0.217 | 0.323 |
| | 720 | 0.357 | 0.430 | 0.372 | 0.449 | 0.398 | 0.444 | 0.360 | 0.434 | 0.347 | 0.311 | 0.342 | 0.388 | 0.358 | 0.461 | 0.211 | 0.317 |
| | Avg | 0.358 | 0.421 | 0.372 | 0.427 | 0.401 | 0.434 | 0.269 | 0.362 | 0.367 | 0.343 | 0.273 | 0.331 | 0.204 | 0.335 | 0.227 | 0.328 |
| From scratch + GPT-Medium + Text prompt | 96 | 0.453 | 0.483 | 0.350 | 0.422 | 0.757 | 0.613 | 0.338 | 0.420 | 0.659 | 0.552 | 0.343 | 0.398 | 0.223 | 0.353 | 0.605 | 0.507 |
| | 192 | 0.422 | 0.470 | 0.348 | 0.423 | 0.708 | 0.601 | 0.326 | 0.415 | 0.509 | 0.475 | 0.323 | 0.388 | 0.212 | 0.352 | 0.352 | 0.403 |
| | 336 | 0.481 | 0.502 | 0.449 | 0.487 | 0.938 | 0.701 | 0.483 | 0.506 | 0.562 | 0.496 | 0.457 | 0.474 | 0.430 | 0.502 | 0.729 | 0.456 |
| | 720 | 0.437 | 0.482 | 0.396 | 0.461 | 0.634 | 0.563 | 0.408 | 0.459 | 0.536 | 0.463 | 0.415 | 0.434 | 0.457 | 0.517 | 0.460 | 0.438 |
| | Avg | 0.448 | 0.484 | 0.385 | 0.448 | 0.759 | 0.619 | 0.389 | 0.450 | 0.566 | 0.496 | 0.384 | 0.423 | 0.330 | 0.431 | 0.537 | 0.451 |
| From scratch + GPT-Small + TS prompt | 96 | 0.323 | 0.392 | 0.243 | 0.341 | 0.394 | 0.437 | 0.185 | 0.301 | 0.334 | 0.321 | 0.200 | 0.279 | 0.098 | 0.236 | 0.197 | 0.291 |
| | 192 | 0.332 | 0.399 | 0.275 | 0.362 | 0.369 | 0.426 | 0.204 | 0.313 | 0.333 | 0.306 | 0.219 | 0.286 | 0.127 | 0.268 | 0.195 | 0.290 |
| | 336 | 0.345 | 0.405 | 0.317 | 0.394 | 0.352 | 0.415 | 0.263 | 0.364 | 0.324 | 0.288 | 0.265 | 0.323 | 0.179 | 0.321 | 0.181 | 0.282 |
| | 720 | 0.362 | 0.432 | 0.364 | 0.447 | 0.389 | 0.439 | 0.324 | 0.402 | 0.341 | 0.300 | 0.339 | 0.377 | 0.333 | 0.457 | 0.207 | 0.309 |
| | Avg | 0.340 | 0.407 | 0.300 | 0.386 | 0.376 | 0.429 | 0.244 | 0.345 | 0.333 | 0.304 | 0.256 | 0.316 | 0.184 | 0.320 | 0.195 | 0.293 |
| Fully tune + GPT-Small + TS prompt | 96 | 0.317 | 0.383 | 0.240 | 0.334 | 0.431 | 0.443 | 0.178 | 0.285 | 0.315 | 0.293 | 0.188 | 0.257 | 0.083 | 0.208 | 0.161 | 0.265 |
| | 192 | 0.355 | 0.413 | 0.285 | 0.370 | 0.479 | 0.482 | 0.221 | 0.323 | 0.352 | 0.336 | 0.238 | 0.304 | 0.132 | 0.277 | 0.213 | 0.311 |
| | 336 | 0.357 | 0.414 | 0.302 | 0.388 | 0.486 | 0.486 | 0.252 | 0.346 | 0.339 | 0.310 | 0.275 | 0.326 | 0.196 | 0.336 | 0.203 | 0.302 |
| | 720 | 0.363 | 0.434 | 0.361 | 0.442 | 0.479 | 0.483 | 0.345 | 0.420 | 0.350 | 0.313 | 0.345 | 0.384 | 0.426 | 0.496 | 0.224 | 0.326 |
| | Avg | 0.348 | 0.411 | 0.297 | 0.384 | 0.469 | 0.473 | 0.249 | 0.343 | 0.339 | 0.313 | 0.261 | 0.317 | 0.209 | 0.329 | 0.200 | 0.301 |
| Fully tune + GPT-Small + Text prompt | 96 | 0.305 | 0.377 | 0.226 | 0.320 | 0.276 | 0.360 | 0.143 | 0.253 | 0.305 | 0.279 | 0.162 | 0.227 | 0.060 | 0.178 | 0.144 | 0.246 |
| | 192 | 0.335 | 0.397 | 0.278 | 0.359 | 0.314 | 0.389 | 0.191 | 0.295 | 0.315 | 0.283 | 0.212 | 0.275 | 0.118 | 0.253 | 0.161 | 0.261 |
| | 336 | 0.348 | 0.406 | 0.310 | 0.392 | 0.344 | 0.411 | 0.239 | 0.339 | 0.323 | 0.285 | 0.266 | 0.318 | 0.198 | 0.333 | 0.175 | 0.277 |
| | 720 | 0.371 | 0.454 | 0.364 | 0.446 | 0.404 | 0.452 | 0.352 | 0.405 | 0.344 | 0.305 | 0.332 | 0.366 | 0.379 | 0.470 | 0.208 | 0.309 |
| | Avg | 0.340 | 0.409 | 0.294 | 0.379 | 0.334 | 0.403 | 0.231 | 0.323 | 0.322 | 0.288 | 0.243 | 0.296 | 0.189 | 0.308 | 0.172 | 0.273 |
| Fully tune + GPT-Medium + Text prompt | 96 | 0.301 | 0.372 | 0.229 | 0.320 | 0.261 | 0.346 | 0.149 | 0.266 | 0.300 | 0.268 | 0.163 | 0.230 | 0.058 | 0.173 | 0.141 | 0.241 |
| | 192 | 0.332 | 0.397 | 0.290 | 0.368 | 0.288 | 0.370 | 0.204 | 0.303 | 0.316 | 0.282 | 0.215 | 0.282 | 0.133 | 0.277 | 0.158 | 0.258 |
| | 336 | 0.351 | 0.412 | 0.316 | 0.392 | 0.343 | 0.413 | 0.294 | 0.376 | 0.328 | 0.295 | 0.281 | 0.332 | 0.224 | 0.369 | 0.175 | 0.276 |
| | 720 | 0.368 | 0.436 | 0.378 | 0.452 | 0.371 | 0.431 | 0.492 | 0.494 | 0.344 | 0.303 | 0.350 | 0.385 | 0.321 | 0.442 | 0.207 | 0.308 |
| | Avg | 0.338 | 0.404 | 0.303 | 0.383 | 0.316 | 0.390 | 0.285 | 0.360 | 0.322 | 0.287 | 0.252 | 0.307 | 0.184 | 0.315 | 0.170 | 0.271 |
| Fully tune + GPT-Medium + Text prompt | 96 | 0.320 | 0.387 | 0.242 | 0.330 | 0.490 | 0.477 | 0.191 | 0.290 | 0.346 | 0.326 | 0.212 | 0.270 | 0.134 | 0.269 | 0.185 | 0.300 |
| | 192 | 0.342 | 0.403 | 0.270 | 0.352 | 0.376 | 0.423 | 0.196 | 0.287 | 0.355 | 0.327 | 0.236 | 0.286 | 0.173 | 0.305 | 0.204 | 0.316 |
| | 336 | 0.348 | 0.409 | 0.284 | 0.367 | 0.530 | 0.501 | 0.253 | 0.335 | 0.379 | 0.345 | 0.298 | 0.331 | 0.311 | 0.421 | 0.224 | 0.334 |
| | 720 | 0.368 | 0.433 | 0.424 | 0.479 | 0.375 | 0.429 | 0.361 | 0.423 | OOM | OOM | OOM | OOM | 0.333 | 0.457 | 0.206 | 0.307 |
| | Avg | 0.344 | 0.408 | 0.305 | 0.382 | 0.443 | 0.458 | 0.250 | 0.334 | 0.360 | 0.333 | 0.249 | 0.295 | 0.238 | 0.363 | 0.205 | 0.314 |
| Fully tune + Phi-2 + TS prompt | 96 | 0.296 | 0.371 | 0.234 | 0.328 | 0.309 | 0.381 | 0.150 | 0.263 | 0.299 | 0.278 | 0.175 | 0.248 | 0.073 | 0.204 | 0.145 | 0.249 |
| | 192 | 0.318 | 0.386 | 0.273 | 0.355 | 0.301 | 0.381 | 0.190 | 0.293 | 0.311 | 0.278 | 0.212 | 0.279 | 0.129 | 0.271 | 0.164 | 0.266 |
| | 336 | 0.337 | 0.402 | 0.311 | 0.389 | 0.346 | 0.419 | 0.283 | 0.381 | 0.323 | 0.290 | 0.282 | 0.345 | 0.233 | 0.374 | 0.179 | 0.281 |
| | 720 | 0.372 | 0.445 | 0.317 | 0.407 | 0.404 | 0.461 | 0.439 | 0.484 | 0.347 | 0.305 | 0.354 | 0.382 | 0.404 | 0.501 | 0.218 | 0.319 |
| | Avg | 0.331 | 0.401 | 0.284 | 0.370 | 0.340 | 0.411 | 0.265 | 0.355 | 0.320 | 0.288 | 0.256 | 0.313 | 0.210 | 0.337 | 0.176 | 0.279 |
| Fully tune + Pi-2 + Text prompt | 96 | 0.296 | 0.371 | 0.234 | 0.328 | 0.309 | 0.381 | 0.150 | 0.263 | 0.299 | 0.278 | 0.175 | 0.248 | 0.073 | 0.204 | 0.145 | 0.249 |
| | 192 | 0.319 | 0.385 | 0.269 | 0.355 | 0.309 | 0.383 | 0.188 | 0.295 | 0.307 | 0.275 | 0.212 | 0.283 | 0.134 | 0.281 | 0.161 | 0.262 |
| | 336 | 0.337 | 0.402 | 0.311 | 0.389 | 0.346 | 0.419 | 0.283 | 0.381 | 0.323 | 0.290 | 0.282 | 0.345 | 0.233 | 0.374 | 0.179 | 0.281 |
| | 720 | 0.356 | 0.430 | 0.359 | 0.442 | 0.392 | 0.454 | 0.383 | 0.451 | 0.345 | 0.302 | 0.345 | 0.377 | 0.561 | 0.606 | 0.212 | 0.315 |
| | Avg | 0.327 | 0.397 | 0.293 | 0.378 | 0.339 | 0.409 | 0.251 | 0.347 | 0.318 | 0.286 | 0.254 | 0.313 | 0.250 | 0.366 | 0.174 | 0.277 |
| LoRA-dim-16 + GPT-Medium + TS prompt | 96 | 0.362 | 0.419 | 0.273 | 0.363 | 0.589 | 0.533 | 0.225 | 0.332 | 0.428 | 0.396 | 0.224 | 0.293 | 0.129 | 0.274 | 0.227 | 0.333 |
| | 192 | 0.394 | 0.444 | 0.312 | 0.397 | 0.582 | 0.531 | 0.259 | 0.361 | 0.502 | 0.437 | 0.339 | 0.280 | 0.200 | 0.345 | 0.257 | 0.358 |
| | 336 | 0.403 | 0.457 | 0.321 | 0.413 | 0.560 | 0.532 | 0.293 | 0.392 | 0.547 | 0.457 | 0.320 | 0.369 | 0.266 | 0.409 | 0.291 | 0.386 |
| | 720 | 0.444 | 0.499 | 0.366 | 0.451 | 0.576 | 0.547 | 0.355 | 0.436 | 0.660 | 0.519 | 0.369 | 0.406 | 0.457 | 0.532 | 0.406 | 0.479 |
| | Avg | 0.401 | 0.455 | 0.318 | 0.406 | 0.577 | 0.536 | 0.283 | 0.380 | 0.534 | 0.452 | 0.313 | 0.337 | 0.263 | 0.390 | 0.295 | 0.389 |
| LoRA-dim-32 + GPT-Medium + TS prompt | 96 | 0.365 | 0.422 | 0.270 | 0.361 | 0.596 | 0.593 | 0.222 | 0.329 | 0.438 | 0.408 | 0.223 | 0.294 | 0.117 | 0.259 | 0.233 | 0.341 |
| | 192 | 0.401 | 0.449 | 0.314 | 0.398 | 0.594 | 0.537 | 0.261 | 0.363 | 0.503 | 0.443 | 0.281 | 0.340 | 0.204 | 0.346 | 0.259 | 0.361 |
| | 336 | 0.403 | 0.457 | 0.321 | 0.413 | 0.563 | 0.533 | 0.294 | 0.393 | 0.547 | 0.459 | 0.321 | 0.370 | 0.267 | 0.410 | 0.294 | 0.390 |
| | 720 | 0.444 | 0.498 | 0.367 | 0.452 | 0.572 | 0.545 | 0.357 | 0.437 | 0.647 | 0.513 | 0.369 | 0.406 | 0.454 | 0.530 | 0.399 | 0.473 |
| | Avg | 0.403 | 0.457 | 0.318 | 0.406 | 0.581 | 0.552 | 0.283 | 0.380 | 0.534 | 0.456 | 0.298 | 0.352 | 0.260 | 0.386 | 0.296 | 0.391 |
| LoRA-dim-16 + GPT-Medium + Word prompt | 96 | 0.377 | 0.431 | 0.284 | 0.376 | 0.603 | 0.538 | 0.239 | 0.348 | 0.462 | 0.423 | 0.244 | 0.313 | 0.154 | 0.302 | 0.242 | 0.348 |
| | 192 | 0.394 | 0.445 | 0.313 | 0.400 | 0.578 | 0.530 | 0.263 | 0.367 | 0.511 | 0.441 | 0.284 | 0.344 | 0.203 | 0.351 | 0.262 | 0.363 |
| | 336 | 0.412 | 0.465 | 0.325 | 0.417 | 0.571 | 0.538 | 0.299 | 0.397 | 0.567 | 0.471 | 0.323 | 0.373 | 0.267 | 0.413 | 0.308 | 0.402 |
| | 720 | 0.448 | 0.501 | 0.368 | 0.452 | 0.582 | 0.550 | 0.357 | 0.437 | 0.672 | 0.526 | 0.370 | 0.407 | 0.452 | 0.529 | 0.416 | 0.486 |
| | Avg | 0.408 | 0.461 | 0.322 | 0.411 | 0.583 | 0.539 | 0.289 | 0.387 | 0.553 | 0.465 | 0.305 | 0.359 | 0.269 | 0.399 | 0.307 | 0.400 |
| LoRA-dim-32 + GPT-Medium + Word prompt | 96 | 0.365 | 0.423 | 0.276 | 0.367 | 0.590 | 0.533 | 0.230 | 0.337 | 0.449 | 0.410 | 0.234 | 0.305 | 0.133 | 0.277 | 0.237 | 0.343 |
| | 192 | 0.400 | 0.449 | 0.311 | 0.397 | 0.572 | 0.527 | 0.261 | 0.364 | 0.515 | 0.447 | 0.284 | 0.345 | 0.207 | 0.352 | 0.265 | 0.366 |
| | 336 | 0.410 | 0.463 | 0.324 | 0.416 | 0.570 | 0.538 | 0.299 | 0.398 | 0.563 | 0.469 | 0.323 | 0.373 | 0.268 | 0.414 | 0.305 | 0.399 |
| | 720 | 0.447 | 0.500 | 0.368 | 0.453 | 0.589 | 0.553 | 0.359 | 0.459 | 0.664 | 0.522 | 0.370 | 0.407 | 0.453 | 0.530 | 0.414 | 0.486 |
| | Avg | 0.406 | 0.459 | 0.320 | 0.408 | 0.580 | 0.538 | 0.287 | 0.389 | 0.547 | 0.462 | 0.303 | 0.357 | 0.265 | 0.393 | 0.305 | 0.398 |

Table 19: Results of different backbones.

| Datasets | | ETTh1 | | ETTh2 | | ETTm1 | | ETTm2 | | Traffic | | Weather | | Exchange | | ECL | |
|---|---|---|---|---|---|---|---|---|---|---|---|---|---|---|---|---|---|
| Metric | | MSE | MAE | MSE | MAE | MSE | MAE | MSE | MAE | MSE | MAE | MSE | MAE | MSE | MAE | MSE | MAE |
| Fully tune | 96 | 0.297 | 0.368 | 0.224 | 0.319 | 0.277 | 0.360 | 0.147 | 0.259 | 0.304 | 0.280 | 0.168 | 0.240 | 0.069 | 0.192 | 0.148 | 0.251 |
| + GPT-Large | 192 | 0.328 | 0.391 | 0.279 | 0.362 | 0.300 | 0.377 | 0.207 | 0.311 | 0.315 | 0.279 | 0.212 | 0.277 | 0.121 | 0.264 | 0.159 | 0.258 |
| + TS prompt | 336 | 0.347 | 0.407 | 0.365 | 0.420 | 0.322 | 0.397 | 0.284 | 0.364 | 0.324 | 0.287 | 0.291 | 0.341 | 0.356 | 0.461 | 0.174 | 0.276 |
| | 720 | 0.358 | 0.427 | 0.430 | 0.478 | 0.358 | 0.421 | 0.436 | 0.467 | 0.348 | 0.303 | 0.370 | 0.388 | 0.424 | 0.525 | 0.207 | 0.308 |
| | Avg | 0.333 | 0.398 | 0.325 | 0.395 | 0.314 | 0.389 | 0.269 | 0.350 | 0.323 | 0.287 | 0.260 | 0.311 | 0.242 | 0.360 | 0.172 | 0.273 |
| Fully tune | 96 | 0.307 | 0.377 | 0.235 | 0.326 | 0.285 | 0.369 | 0.156 | 0.266 | 0.305 | 0.278 | 0.172 | 0.242 | 0.065 | 0.186 | 0.145 | 0.247 |
| + GPT-Medium | 192 | 0.329 | 0.391 | 0.283 | 0.365 | 0.319 | 0.393 | 0.203 | 0.307 | 0.313 | 0.274 | 0.218 | 0.248 | 0.115 | 0.248 | 0.159 | 0.259 |
| + TS prompt | 336 | 0.346 | 0.405 | 0.320 | 0.401 | 0.378 | 0.425 | 0.255 | 0.349 | 0.326 | 0.287 | 0.276 | 0.329 | 0.206 | 0.339 | 0.180 | 0.284 |
| | 720 | 0.370 | 0.441 | 0.378 | 0.456 | 0.464 | 0.477 | 0.342 | 0.417 | 0.346 | 0.301 | 0.339 | 0.373 | 0.409 | 0.487 | 0.215 | 0.317 |
| | Avg | 0.338 | 0.403 | 0.304 | 0.387 | 0.362 | 0.416 | 0.239 | 0.335 | 0.323 | 0.285 | 0.251 | 0.305 | 0.199 | 0.315 | 0.175 | 0.276 |
| Fully tune | 96 | 0.317 | 0.383 | 0.240 | 0.334 | 0.431 | 0.443 | 0.178 | 0.285 | 0.315 | 0.293 | 0.188 | 0.257 | 0.083 | 0.208 | 0.161 | 0.265 |
| + GPT-Small | 192 | 0.355 | 0.413 | 0.285 | 0.370 | 0.479 | 0.482 | 0.221 | 0.323 | 0.352 | 0.336 | 0.238 | 0.304 | 0.132 | 0.277 | 0.213 | 0.311 |
| + TS prompt | 336 | 0.357 | 0.414 | 0.302 | 0.388 | 0.486 | 0.486 | 0.252 | 0.346 | 0.339 | 0.310 | 0.275 | 0.326 | 0.196 | 0.336 | 0.203 | 0.302 |
| | 720 | 0.363 | 0.434 | 0.361 | 0.442 | 0.479 | 0.483 | 0.345 | 0.420 | 0.350 | 0.313 | 0.345 | 0.384 | 0.426 | 0.496 | 0.224 | 0.326 |
| | Avg | 0.348 | 0.411 | 0.297 | 0.384 | 0.469 | 0.473 | 0.249 | 0.343 | 0.339 | 0.313 | 0.261 | 0.317 | 0.209 | 0.329 | 0.200 | 0.301 |
| Fully tune | 96 | 0.296 | 0.371 | 0.234 | 0.328 | 0.309 | 0.381 | 0.150 | 0.263 | 0.299 | 0.278 | 0.175 | 0.248 | 0.073 | 0.204 | 0.145 | 0.249 |
| + Phi-2 | 192 | 0.318 | 0.386 | 0.273 | 0.355 | 0.301 | 0.381 | 0.190 | 0.293 | 0.311 | 0.278 | 0.212 | 0.279 | 0.129 | 0.271 | 0.164 | 0.266 |
| + TS prompt | 336 | 0.337 | 0.402 | 0.311 | 0.389 | 0.346 | 0.419 | 0.283 | 0.381 | 0.323 | 0.290 | 0.282 | 0.345 | 0.233 | 0.374 | 0.179 | 0.281 |
| | 720 | 0.372 | 0.445 | 0.317 | 0.407 | 0.404 | 0.461 | 0.439 | 0.484 | 0.347 | 0.305 | 0.354 | 0.382 | 0.404 | 0.501 | 0.218 | 0.319 |
| | Avg | 0.331 | 0.401 | 0.284 | 0.370 | 0.340 | 0.411 | 0.265 | 0.355 | 0.320 | 0.288 | 0.256 | 0.313 | 0.210 | 0.337 | 0.176 | 0.279 |

Table 20: Results of different down-sampling ratios. Experiments with GPT-Medium as backbones, TS prompt, and fully tuning paradigm.

| Datasets | | ETTh1 | | ETTh2 | | ETTm1 | | ETTm2 | | Traffic | | Weather | | ECL | |
|---|---|---|---|---|---|---|---|---|---|---|---|---|---|---|---|---|
| Metric | | MSE | MAE | MSE | MAE | MSE | MAE | MSE | MAE | MSE | MAE | MSE | MAE | MSE | MAE |
| Downsample Ratio = 40 | 96 | 0.4536 | 0.4787 | 0.3395 | 0.4097 | 0.7441 | 0.6068 | 0.3229 | 0.4055 | 0.6619 | 0.5636 | 0.3538 | 0.3979 | 0.6118 | 0.499 |
| | 192 | 0.4648 | 0.4935 | 0.386 | 0.4507 | 0.7659 | 0.6336 | 0.3797 | 0.4604 | 0.6644 | 0.5537 | 0.4106 | 0.454 | 0.6756 | 0.4915 |
| | 336 | 0.6629 | 0.6167 | 0.5982 | 0.5916 | 1.1366 | 0.8212 | 0.6904 | 0.6455 | 1.0706 | 0.7785 | 0.7359 | 0.65 | 1.6485 | 0.7188 |
| | 720 | 1.0518 | 0.8133 | 0.8802 | 0.7588 | 1.8609 | 1.1304 | 1.2011 | 0.9073 | 1.7276 | 1.062 | 1.3176 | 0.9268 | 3.9988 | 1.0223 |
| | Avg | 0.343 | 0.408 | 0.307 | 0.390 | 0.353 | 0.412 | 0.234 | 0.331 | 0.344 | 0.314 | 0.253 | 0.306 | 0.210 | 0.330 |
| Downsample Ratio = 20 | 96 | 0.307 | 0.3767 | 0.2349 | 0.3263 | 0.285 | 0.3687 | 0.1555 | 0.2657 | 0.3053 | 0.2778 | 0.1717 | 0.2416 | 0.1447 | 0.2468 |
| | 192 | 0.3288 | 0.3908 | 0.2826 | 0.3649 | 0.3193 | 0.3925 | 0.2032 | 0.3069 | 0.3132 | 0.2744 | 0.2177 | 0.2782 | 0.1587 | 0.2588 |
| | 336 | 0.346 | 0.405 | 0.3198 | 0.4005 | 0.3782 | 0.4254 | 0.2549 | 0.3492 | 0.3263 | 0.2869 | 0.2761 | 0.3287 | 0.1803 | 0.2835 |
| | 720 | 0.3704 | 0.4405 | 0.378 | 0.456 | 0.4638 | 0.4773 | 0.3422 | 0.4172 | 0.3456 | 0.301 | 0.3386 | 0.3732 | 0.2151 | 0.3167 |
| | Avg | 0.338 | 0.404 | 0.303 | 0.383 | 0.316 | 0.390 | 0.285 | 0.360 | 0.322 | 0.287 | 0.252 | 0.307 | 0.184 | 0.315 |
| Downsample Ratio = 10 | 96 | 0.2975 | 0.3698 | 0.2268 | 0.3175 | 0.2633 | 0.3462 | 0.1463 | 0.2583 | 0.2961 | 0.2626 | 0.2583 | 0.2247 | 0.1406 | 0.2411 |
| | 192 | 0.3293 | 0.3896 | 0.2848 | 0.3674 | 0.3286 | 0.3991 | 0.1995 | 0.3014 | 0.3089 | 2673 | 0.2151 | 0.2786 | 0.1568 | 0.2564 |
| | 336 | 0.3461 | 0.4039 | 0.3097 | 0.3938 | 0.3593 | 0.4198 | 0.259 | 0.3505 | 0.3206 | 0.2785 | 0.2651 | 0.3155 | 0.1744 | 0.2747 |
| | 720 | 0.3676 | 0.4333 | 0.4101 | 0.4738 | 0.4096 | 0.4505 | 0.3632 | 0.4242 | 0.3428 | 0.2983 | 0.3462 | 0.38 | 0.2099 | 0.3101 |
| | Avg | 0.335 | 0.402 | 0.321 | 0.392 | 0.341 | 0.399 | 0.235 | 0.332 | 0.312 | 0.274 | 0.252 | 0.308 | 0.232 | 0.360 |

