# OpenReview forum: "LTSM-Bundle: A Toolbox and Benchmark on Large Language Models for Time Series Forecasting"
_TMLR — Rejected by TMLR_

### Review · Reviewer_mkxF · 2025-08-19

**Summary Of Contributions:**

This work presents an experimental evaluation of various techniques applied in LLM-based time series forecasting methods, including prompt design, tokenization approches, training paradigms, base model selectoin, data quantity and data diversity. Building on these findings, the authors propose to combine the best performed techniques together to form a large time series model. The resulting model is shown to perform well on several benchmark datasets.

Strength:
- The writing of the paper is clear and easy to follow.

Weaknesses:
- Lack of structured review: The paper does not provide a structured or systematic review of existing LLM-based time series forecasting methods. As a result, its scholarly contribution is limited, since readers gain little understanding of how this work fits into or advances the broader research landscape.

- Superficial treatment of techniques: The techniques evaluated are introduced only briefly, without sufficient explanation or analysis of their underlying mechanisms. This lack of depth prevents the reader from understanding why these methods may succeed or fail, and misses the opportunity to provide valuable theoretical or methodological insights.

- Insufficient analysis of results: The experimental findings are presented without meaningful interpretation or critical discussion. The absence of deeper analysis makes the work resemble an experimental report rather than a research paper, reducing its academic significance and impact.

**Audience:**

No

**Audience Explanation:**

As outlined in the Strengths and Weaknesses section, the current version of this work does not provide sufficient insights or contributions to be of significant interest to the community. While the empirical evaluation has potential value, the lack of in-depth analysis, theoretical framing, and critical discussion limits its relevance and impact. I encourage the authors to strengthen the paper by providing deeper analysis of the evaluated techniques, clearer interpretation of the results, and more critical reflection on the implications for future research.

**Broader Impact Concerns:**

No broader impact concerns.

**Claims And Evidence:**

No

**Claims Explanation:**

The paper claims that combining the best-performing techniques leads to state-of-the-art performance on benchmark datasets. However, this claim is not convincingly supported. In particular, the zero-shot and few-shot experimental settings are insufficiently described, leaving ambiguity about how the experiments were designed and how fair the comparisons are. Without clear definitions of these settings and a transparent explanation of the evaluation protocol, the reported results cannot be fully trusted or independently verified.

**Requested Changes:**

Please address the weaknesses in the Strengths and Weaknesses section. The three weaknesses points represent my major concerns regarding the acceptance of this paper.

---

### Review · Reviewer_N9Cw · 2025-08-21

**Summary Of Contributions:**

The paper proposes a pipeline for the training and evaluation of time series models and provides a library to do so. The paper then performs a series of ablations on different aspects of their proposed pipeline. Eventually, the paper presents the best ablated model on the LSF benchmark with non pre-trained baselines.

**Audience:**

No

**Audience Explanation:**

The paper is highly confusing, and is not consistently written nor does it have findings that it claims it does. The abstract and introduction makes many references to Large Time Series Models, but is operating in the older paradigm, focusing on the long sequence forecasting datasets.

**Broader Impact Concerns:**

None.

**Claims And Evidence:**

No

**Claims Explanation:**

The paper is highly overstated. It makes references to Large Time Series Models such as Chronos, TimesFM, Moirai, which are pre-trained models with zero-shot forecasting capabilities. However, in practice, the paper is tackling the typical non pre-trained setting on the long sequence forecasting benchmark. It is quite confusing why the paper keeps making reference to LTSMs and makes many claims of being a general library when it is clearly not. Even figure 1 is quite confusing, where the figure prominently features an LLM, rather than a general sequence model.

It is lacking many features to be a library to investigate different aspects of modern LTSMs. There have been many variations of LTSMs recently, does it support autoregressive/decoder-only models? It also does not seem to support probabilistic forecasts. Comparisons are not made against other TSFMs, but only compares with some outdated models.

There are also some inconsistencies, such as claims of supporting multiple loss functions, but other parts of the paper say that they only support MSE.

It is also unclear what are the benefits of this library over other existing libraries, such as THUML's time-series-library, gluonts, probts, and many others. Taking a look at the codebase, many parts are in fact lifted from THUML time-series-library.

**Requested Changes:**

Major updates need to be made, and can be done in 2 ways.

The first would be to not overstate the goals of the work, removing mentions about LTSMs.

The other would be to update the library and to include many of the recent design choices, datasets (e.g. LOTSA, UTSD, Chronos dataset, etc.) and compare to these TSFMs.

---

> ### Author Response · Authors · 2025-09-02
> **Response to Reviewer N9Cw**
>
> **[Q-4]** It is also unclear what are the benefits of this library over other existing libraries, such as THUML's time-series-library, gluonts, probts, and many others.
>
> **[A-4]** We thank the reviewer for raising this important point. Here, we clarify the unique contributions of LTSM-Bundle relative to existing libraries such as THUML’s time-series-library, GluonTS, and ProbTS.
>
> First, while LTSM-Bundle adapts a few low-level components (e.g., data loaders and metric utilities) from THUML’s time-series-library to ensure consistency with common benchmarks, the core contributions are original:
> - Unified benchmarking for modern LTSMs and LLM-based models, unlike THUML and GluonTS, which focus mainly on traditional forecasting architectures, LTSM-Bundle supports decoder-only models and integrates pre-trained LLMs such as Phi-2 (results in Table 3).
> - Modular tokenization strategies. We standardize linear vs. time-series tokenization across models, enabling cross-architecture comparability (Section 3.2).
> - Zero-shot and few-shot evaluation protocols. The framework provides standardized pipelines for evaluating pre-trained LTSMs (e.g., Chronos, Moirai) alongside conventional supervised models (Section 4).
> - Extensible model integration. The library is designed to seamlessly incorporate new LTSMs, probabilistic metrics, and hybrid architectures (Section 2.3).
>
> Second, compared with GluonTS and ProbTS, LTSM-Bundle focuses specifically on LLM-driven and large time series models while providing standardized experimental settings for long-sequence forecasting, a capability missing in existing libraries.
>
> Our implementation does not reuse code from THUML. Instead, as noted in Section 3.1 and the supplementary material, we refer to the structural design of GPT4TS as an initial reference point, but redesigned the framework extensively to support benchmarking the training component and to handle diverse LTSM and LLM-based models. Specifically, while GPT4TS primarily focuses on inference-only pipelines, LTSM-Bundle introduces a new, modular benchmarking framework that:
> - Redesigns the training pipeline to support full fine-tuning, LoRA, and zero-/few-shot evaluation under a unified interface (Section 4).
> - Integrates decoder-only LLMs such as Phi-2 (results in Table 3) and enables benchmarking of emerging pre-trained LTSMs like Chronos and Moirai.
> Implements standardized tokenization strategies (linear vs. time-series) for fair cross-model comparisons (Section 3.3).
> - Establishes consistent evaluation protocols across 16 models and 15 datasets, including both supervised and pre-trained zero-shot forecasting settings.
>
> **[Q-5]** The paper is highly confusing, and is not consistently written nor does it have findings that it claims it does. The abstract and introduction makes many references to Large Time Series Models, but is operating in the older paradigm, focusing on the long sequence forecasting datasets.
>
> **[A-5]** We thank the reviewer for mentioning that the current phrasing in the abstract and introduction may have caused confusion regarding the scope of the paper. To clarify, **the goal of LTSM-Bundle is to provide a unified benchmarking framework** that evaluates both:
> - Traditional long-sequence forecasting (LSF) models (e.g., PatchTST, FEDformer, TimesNet), and
> - Emerging Large Time Series Models (LTSMs) and LLM-based approaches with zero-shot and few-shot capabilities (e.g., Phi-2, Chronos, Moirai).
>
> While our evaluation leverages widely used long-sequence forecasting datasets to ensure comparability with existing benchmarks, we also explicitly include LTSM-style models in the experiments — for example, Phi-2 (decoder-only LLM), with results reported in Table 3 and Figure 5. These evaluations demonstrate key findings, including:
> - Tokenization strategies significantly affect cross-model transferability (Section 5.2).
> - Smaller or medium-sized models can match or outperform larger LLMs on long-horizon forecasting (Section 5.3).
> - Zero-shot and few-shot paradigms provide competitive baselines for integrating pre-trained LTSMs into forecasting pipelines (Section 5.4).
>
> To improve clarity and consistency, we will revise the abstract and introduction to:
> - Better distinguish traditional LSF models from pre-trained LTSMs.
> - Clearly state that LTSM-Bundle bridges both paradigms rather than focusing exclusively on one.
> - Explicitly summarize our findings to avoid any ambiguity about the paper’s contributions.

---

### Review · Reviewer_msTi · 2025-08-24

**Summary Of Contributions:**

The paper introduces a toolbox and benchmark for Large Time Series Models (LTSMs) with a user‑friendly interface. It systematically investigates the impact of various components, including prompting strategies, tokenization methods, training paradigms, base model selection, data quantity, and dataset diversity, on the overall model performance. Through extensive experiments, the paper presents observations on the performance of the GPT‑2 model on time series datasets and compares the best-performing configuration with state-of-the-art baselines.

**Audience:**

Yes

**Audience Explanation:**

1. The paper addresses a practical need in LTSMs by providing a more unified and comprehensive interface for handling diverse datasets.
2. The proposed toolbox supports multiple model architectures and prompting strategies, offering a user-friendly interface that allows users to conveniently explore and experiment with different configurations.

**Broader Impact Concerns:**

No concerns.

**Claims And Evidence:**

No

**Claims Explanation:**

1. The impact of each components, including prompting strategies, tokenization methods, training paradigms, base model selection, data quantity, and dataset diversity is evaluated independent and individually which may rise issues about the interactions among these components.
2. Although the paper positions itself as a toolbox and benchmark, it primarily focuses on introducing a prediction function, making it appear more like a single-method contribution rather than a comprehensive toolbox. It would be beneficial to include demonstrations of other functions highlighted in Table 13 and Fig.2 and briefly present their results in the Appendix to better showcase the capabilities of the interface. In addition, the range of evaluated datasets is relatively limited. For instance, datasets such as ILI and Exchange are missing from Table 9. There are also additional datasets available that have been used in other recent LTSM papers and could be leveraged to strengthen the evaluation.
3. In the evaluation, the proposed method is tested using the best configurations identified across different modules. It is unclear whether the baseline methods are also evaluated under their optimal configurations. If the original baseline papers do not clearly specify the best settings for down sample rate, it may be more accurate to evaluate baselines using the full datasets rather than a restricted 5% rate, since the latter is applied only to the proposed method and may create an unfair comparison.
4. It is unclear what the key novelty or contribution of the proposed method is beyond integration. Could the authors clarify what fundamentally differentiates the method and explain what makes it outperforms other state-of-the-art baselines?

**Requested Changes:**

1. Please refer to the previous comments.
2. In the Pre-processing: Instruction Prompts section on page 5, there are two identical paragraphs.
3. In Table 6, it would be helpful to also include results from training on the full dataset without down-sampling.
4. Please bold the best results in the Appendix tables as well.

---

### Decision · Action_Editor_HPcA · 2025-09-26

**Recommendation:** Reject

**Audience:**

No

**Audience Explanation:**

While Large Time Series Models (LTSMs) are of interest to the general TMLR community, the reviewers found the models and experiments supported/evaluated to be outdated and the supported library features extremely narrow, thus limiting potential interest from TMLR's audience.  Given the significantly wider scope and feature support of other existing LTSM packages, and that the presented toolbox heavily borrows code from existing LTSM packages, the contributions of the current presented software and experiments remain limited.

**Claims And Evidence:**

No

**Claims Explanation:**

Reviewers cited major concerns with the claims made versus the actual features and models supported via the toolbox.  While the authors responded to these reviewer criticisms, the proposed fixes are described as future additions to the software.  With limited scientific contributions, I would suggest the authors resubmit once fully addressing the reviewers' detailed criticisms within supported toolbox features, demonstrated experiments, and the presented experimental designs.

**Resubmission Of Major Revision:**

The authors may consider submitting a major revision at a later time.